# An enhancement to sea ice motion and age products at NSIDC

Mark A. Tschudi[1], Walter N. Meier[2], J. Scott Stewart[2]

[1]CCAR, Department of Aerospace Engineering Sciences, University of Colorado-Boulder, UCB 431, Boulder, CO, USA, 80309

[2]National Snow and Ice Data Center, CIRES, University of Colorado-Boulder, UCB 449, Boulder, CO, USA, 80309

*Correspondence to*: Mark A. Tschudi (mark.tschudi@colorado.edu)

**Abstract.** A new version of sea ice motion and age products includes several significant upgrades in processing, corrects known issues with the previous version, and updates the time series through 2018,

with regular updates planned for the future. First, we provide a history of these NASA products distributed at the National Snow and Ice Data Center. Then we discuss the improvements to the algorithms, provide validation results for the new (Version 4) and older versions and intercompare the two. While Version 4 algorithm changes were significant, the impact on the products is relatively minor, particularly for more recent years. The changes in Version 4 reduce motion biases by ~0.01 to 0.02 cm/s and error standard

deviations by ~0.3 cm/s. Overall, ice speed increased in Version 4 over Version 3 by 0.5 to 2.0 cm/s over most of the time series. Version 4 shows a higher positive trend for the Arctic of 0.21 cm/s/decade compared to 0.13 cm/s/decade for Version 3. The new version of ice age estimates indicates more older ice than Version 3, especially earlier in the record, but similar trends toward less multi-year ice. Changes in sea ice motion and age derived from the product show a significant shift in the Arctic ice cover, from

a pack with a high concentration of older ice, to a sea ice cover dominated by first-year ice, which is more susceptible to summer melt. We also observe an increase in the speed of the ice over the 30+ year time series, which has been shown in other studies and is anticipated with the annual decrease in sea ice extent.

## 1 Introduction

Arctic sea ice conditions have undergone significant changes in recent years with dramatic reductions in

the overall ice extent, ice age, and ice thickness. The decline in Arctic sea ice extent is one of the better-known and more striking examples of a changing Arctic [e.g. *Meier et al.*, 2014; *Comiso et al.*, 2008;

2012; 2017a; *Stroeve et al.*, 2011; 2014]. Recent estimates indicate that September Arctic sea ice extent has decreased by approximately 13% per decade since 1979, with record or near-record minimum extents occurring several times in the last few years [e.g., *Perovich et al.*, 2019]. In the Antarctic, the trends are smaller and there is higher interannual variability [e.g., *Parkinson and Cavalieri*, 2012]; overall the

Antarctic trends are slightly positive, but with strong regional variability [*Comiso et al.*, 2017b].

Data on sea ice thickness are far less comprehensive and it is more difficult to determine solid quantitative thickness or volume trends. However, there is broad evidence, from observations [e.g., *Kwok*, 2018] and models [e.g., *Stroeve et al.,* 2014] that Arctic sea ice thinning trends are even stronger than the extent

decrease. One explanation for the stronger decline in ice thickness is the preferential loss of thicker, old ice in comparison with relatively thin first-year ice. For example, *Johannessen et al.* [1999] and *Comiso et al.* [2008, 2012] noted the decline in multiyear sea-ice was roughly twice that of first-year ice. In a study examining ice age since the early 1980s, *Maslanik et al.* [2011] found continued recent loss of the oldest ice types, which accelerated starting in 2005. This trend has continued through 2019 [*Perovich et*

*al.*, 2019; *Kwok*, 2018].

A continued decline in the sea-ice cover and the shift from thick multiyear ice (MYI) to more easily navigable first-year ice (FYI) arguably will have one of the biggest impacts on humans and the Arctic environment [e.g., *Pizzolato et al.*, 2016]. In particular, the prospect of new shipping lanes, extraction of

oil and gas from previously inaccessible regions, and increased national security concerns associated with easier and more accessible Arctic waters have already been identified as significant economic and cultural changes related to the sea-ice cover [*Huntington et al.*, 2007]. More open water along the coast will also add to the risk of storm surge and coastal erosion [*Vermaire et al.*, 2013; *Francis et al.*, 2006, 2005; *Lynch et al.,* 2004] and there is some evidence that reductions in sea ice may affect locations far from the Arctic

[e.g., *Overland*, 2016], manifesting particularly through extreme weather in the mid-latitudes [e.g., *Cohen at al.*, 2014; *Francis and Vavrus*, 2012].

The distribution of the age of Arctic sea ice contributes to the vulnerability of the ice cover during the melt season because older ice is on average thicker than younger ice [*Maykut* , 1986; *Tucker et al.,* 2001; *Yu et al.*, 2004; *Tschudi et al.*, 2016], at least in terms of thermodynamic growth over several years. Younger ice is more susceptible to deformation and melts out more readily during the summer, whereas older ice is more likely to remain through the melt season if it does not advect out of the Arctic Ocean. However, the thickness increase with age diminishes over time so that as sea ice gets older [*Maslanik et al.*, 2011], its resiliency against melt does not continually increase. The pre-melt distribution of ice age may therefore serve as a descriptive predictor of how much sea ice will disappear during the melt season and indicate where summer ice loss is more likely to occur.

Ice thickness observations are becoming more widely and readily available from satellite altimeters such as NASA's Ice, Cloud, and land Elevation Satellite (ICESat) [*Kwok and Cunningham*, 2008] and ESA's CryoSat-2 [*Laxon et al*., 2013; *Kurtz et al*., 2014]. However, these satellite-derived data cover a limited time span. ICESat collected twice-yearly estimates from 2003 to 2008 and CryoSat-2, launched in 2010, produces complete Arctic-wide fields monthly [*Tilling et al*., 2016]. The laser altimeter on NASA ICESat-2, launched in September 2018 [*Markus et al*., 2017], provides a significant new source of snow+ice freeboard, and potentially thickness. Along with satellite-borne altimeters, NASA's Operation IceBridge has yielded thickness estimates during 2009 through 2019 in selected regions [e.g., *Kurtz et al*., 2013]. Submarine upward-looking sonar has also been used to estimate thickness sporadically since the 1950s over a selected region in the central Arctic. These data have been connected to the satellite altimetry record [*Kwok*, 2018] to create an intermittent long-term timeseries over part of the Arctic. While these direct ice thickness estimates are useful, such products lack the long-term and/or the basin-wide coverage that is available from the multi-decadal sea ice age record.

In contrast, sea ice motion can be used to track parcels in a Lagrangian sense and record their age. Several sea ice motion products have been developed by various groups. Most products use some sort of motion tracking approach to estimate the drift of features or patterns in satellite images. The EUMETSAT Ocean and Sea Ice Satellite Application Facility (OSI SAF) has two products. One is a low-resolution (62.5 km

spacing) product that derives 2-day motions based on passive microwave and scatterometer inputs [*Lavergne et al.*, 2010]. A medium resolution OSI SAF product based on visible/infrared sensor inputs provides daily coverage at 20 km spatial resolution [*Dybkjaer*, 2018]. Another product, developed by the French National Institute for Ocean Science (IFREMER), combines passive microwave and scatterometer inputs to produce 3-day motion estimates [*Girard-Arduin and Ezraty*, 2012]. Several of these products were inter-compared in *Sumata et al.* [2014, 2015]. High resolution SAR imagery has also been used to track motion at much finer spatial scales [e.g., Curlander et al., 1985; Kwok et al., 2003; Howell et al., 2018]. While high resolution, SAR has had limited spatial and temporal coverage, the data were large and difficult to work with, and reasonable coverage did not start until the mid-1990s. This has changed in recent years, but long-term climate records from SAR are limited.

In this paper, we specifically discuss the "Polar Pathfinder Daily 25 km EASE-Grid Sea Ice Motion Vectors" product [*Tschudi et al.*, 2019a]. These sea ice motions derived from satellite instruments and buoys are then used to obtain a continuous, complete, long-term record of sea ice age, the "EASE-grid Sea Ice Age" product [*Tschudi et al.*, 2019b]. Because of its length and completeness, this ice age timeseries has been used in several studies [*Maslanik et al.*, 2007, 2011; *Tschudi et al.*, 2016; 2010] and reviews of Arctic change [*Stroeve et al.*, 2011; *Meier et al.*, 2014; *Perovich et al.*, 2019] to assess changes in the ice cover. Over time, enhancements and improvements have been made to the ice motion and ice age products. The latest version of the ice motion and age products addresses issues noted by users [*Szanyi et al.*, 2016] and both products are enhanced through a refined optimal interpolation approach that improves the spatial continuity of the gridded motion and age fields. Our focus in this paper is to highlight the changes in the new version, compare the new version with older versions, and provide an updated assessment of ice age trends. As further background, we also document the algorithms and production of the products. Because the ice age product is produced by utilizing the sea ice motion product, we outline the production of the motion product first.

## 2 The Polar Pathfinder Sea Ice Motion Product

The sea ice motion product is archived and distributed by the NASA Snow and Ice Distributed Active Archive Center (DAAC) at the National Snow and Ice Data Center (NSIDC). The ice motion product provides gridded daily estimates and weekly averages of ice motions for both the Arctic and Antarctic regions. In this section, we describe the basic processing methodology and data sources, as well as noting the changes made in the new Version 4 of the product. The version history of the motion product (and the age product discussed in Section 3) is summarized in Table 1, including the release date and enhancements for each version.

### 2.1 Sea ice motion data sources and derivation techniques

Here we provide an overview of the source data and the basic derivation approach. Further details are provided in the product User Guide, available at NSIDC (https://nsidc.org/data/nsidc-0116). There are three primary types of sources for the sea ice motion product: (1) gridded satellite imagery – from several sources, (2) winds from reanalysis fields, and (3) buoy position data. Motions are independently derived from each of these sources. A list of the sources, temporal coverage, and spatial resolution is provided in Table 2. A complete daily gridded product is then produced by combining all sources via an optimal interpolation scheme (Figure 1), which is described further below.

*Gridded satellite imagery*

The approach used for deriving ice motion from satellite imagery is a pattern-matching method that uses cross correlations between patterns in coincident images separated by a given time interval. Such an approach is commonly called "feature-tracking", but at the spatial scales for these images, it is a spatial pattern of many features that are being tracked. Specifically, for our product, motion vectors are computed using a maximum cross-correlation (MCC) pattern-matching method [*Emery et al.*, 1991, 1995]. Two geolocated, spatially-coincident, temporally-consecutive satellite images are selected. Typical time separation between images is 1 to 3 days. For each valid sea ice grid cell, a "search window"

is defined for a region around that grid cell, sized so that it will encompass the range of potential motion during the prescribed time interval (typically ~50 km beyond the grid cell in all directions). The later image is translated relative to the earlier image within this search window, and the correlation between the two images is calculated for each translation. The highest correlation value, i.e., the correlation peak,

5    is assumed to coincide with the most likely offset in the position of the grid cell between the earlier and the later image. This offset in the position yields a displacement vector pointing into the direction of the ice motion; the ice velocity is computed by dividing its magnitude by the time separation between the two images used. All satellite motions are calculated as $u$ and $v$ vector components relative to the EASE grid employed for the product.

The imagery sources have changed over time, depending on which inputs have been available. The primary source has been passive microwave imagery from a series of sensors. Horizontal and vertical polarization fields of 37 GHz and 85/91 GHz channels are used when available. These began in late 1978 with the Scanning Multichannel Microwave Radiometer (SMMR) on the NASA Nimbus-7 platform,

which operated until August 1987 (SMMR did not include the 85 GHz channels). After SMMR, a series of Special Sensor Microwave Imagers (SSMI) on U.S. Defense Meteorological Satellite Program (DMSP) platforms carried on the time series. These were used for the motion product through 2006. Starting in 2007, the motion product transition to the DMSP successor instrument, the Special Sensor Microwave Imager and Sounder (SSMIS), of which three still continue to operate (as of March 2020). The SSMI and

SSMIS imagery are derived from the DMSP SSM/I-SSMIS Daily Polar Gridded Brightness Temperatures, Version 4 product [*Maslanik and Stroeve*, 2004] and the SMMR imagery are from the Nimbus-7 SMMR Polar Gridded Radiances and Sea Ice Concentrations, Version 1 product [*Gloersen*, 2006]. Motions were derived from both the 37 GHz and 85/91 GHz channels from SSMI and SSMIS.

These SMMR-SSMI-SSMIS sources are useful because they provide complete daily (every other day for SMMR) coverage in all-sky conditions (i.e., including night and through clouds). However, their low spatial resolution limits the resolution of motion estimates that can be retrieved. For SMMR, SSMI, and SSMIS the 37 GHz fields are gridded at 25 km resolution, while 85/91 GHz fields are gridded at 12.5 km

resolution. However, the actual resolution, i.e., the sensor footprint, is even coarser; so the effective resolution of the imagery is lower than the gridded resolution. For the SSMI-SSMIS fields, with a gridded resolution of 25 km, daily velocity can only be estimated to the nearest 25 km/day for each velocity component (and actually less in terms of the sensor footprint resolution). This result in a coarse and noisy

motion field. For this reason, similar motion-tracking methods reduce the effect of the coarseness though interpolating the cross-correlation function [e.g., *Kwok et al.*, 1998] or through continuous optimization methods [*Lavergne et al.*, 2010]; often other methods also use a two or three day time separation to reduce noise. Our product obtains useful daily motions by applying an oversampling procedure – effectively moving the correlation window fractions of grid cells – to obtain sub-pixel resolution. During initial

development of the motion algorithm, various oversampling intervals were evaluated for improvement in accuracy versus computational expense. Based on these empirical analyses, an oversampling of 4X was chosen. This oversampling is applied to all satellite estimates. This improves the SSMI-SSMIS effective sampling interval to 6.25 km/day, which corresponds to a theoretical motion precision of 7.23 cm/s. The optimal interpolation method described below smooths this "discretized" motion, allowing estimation of

much slower motions.

In 2002, a more advanced passive microwave sensor, the NASA/JAXA Advanced Microwave Scanning Radiometer for the Earth Observing System (AMSR-E), was launched on the NASA Aqua satellite and operated until October 2011. AMSR-E has more than double the spatial resolution of SSMI/SSMIS.

AMSR-E 89 GHz data are gridded at 6.25 km resolution, compared to 12.5 km for 85/91 GHz SSMI/SSMIS; likewise, AMSR-E 36 GHz data are gridded to 12.5 km compared to 25 km for 37 GHz SSMI/SSMIS. With the higher resolution of the source data, AMSR-E's motion resolution is likewise improved. So, during this period (2002-2011), brightness temperatures from AMSR-E [Cavalieri et al., 2014a,b] were also used as a source for ice motions in the Northern Hemisphere. In 2012, JAXA launched

AMSR2 on their Global Change Observation Mission – Water (GCOM-W) satellite, which continues to operate (as of March 2020). AMSR2 has not yet been added as a source, but this is planned for a future release of the motion product.

For the period, 1981-2000, vectors were produced from the Advanced Very High Resolution Radiometer (AVHRR) for the Northern Hemisphere. AVHRR is a visible/infrared sensor that provides higher spatial resolution than the passive microwave sources. Daily gridded composites at 4 km resolution were used as input to the maximum cross-correlation algorithm [*Emery et al.*, 2000]. The higher resolution of the sensor provided more precise motion estimates than the SMMR-SSMI source. However, motions could only be derived when there were cloud-free conditions on consecutive days. This yielded relatively few vectors, and the impact of AVHRR on the gridded composite fields was relatively small. The AMSR-E 89 GHz channel nearly matches the AVHRR gridded resolution. While the 89 GHz channels are affected by atmospheric emission, retrievals through many cloud conditions are possible, which allows AMSR-E to obtain many more valid motion estimates than AVHRR, at a comparable spatial scale. In addition, the 37 GHz channels have less atmospheric emission and while lower resolution still mark a substantial improvement over SSMI and SSMIS. Thus, inclusion of AVHRR as a motion source was discontinued after 2000 (when the source AVHRR product ended).

To further reduce errors, post-processing filtering techniques are applied to the cross-correlation scheme. First, a minimum correlation threshold of 0.4 is applied to the motion estimates from all of the satellite-derived MCC estimates. This removes 'weak' matches that are more likely to be incorrect. Various thresholds were investigated during the original development of the method [*Emery et al.*, 1991] and 0.4 was determined to be reasonable in terms of balancing the allowance of too many erroneous matches versus incorrectly removing many "good" matches [*Emery et al.*, 1986]. Our value of 0.4 is a subjective choice, but is within the range of thresholds chosen by other methods, e.g.: 0.6 [*Girard-Ardhuin and Ezraty*, 2012] or 0.3 [*Kwok et al.*, 1998; Lavergne et al., 2010].

Second, a neighborhood filter is applied to each individual motion source. At the low-resolution of the satellite data, motion is spatially well-correlated across several grid cells. For each vector retrieved, it is compared with two neighboring vectors. To pass the filter, the motion displacement must be consistent within two grid cells of the displacements of the two neighboring vectors. If the displacements are not consistent within the 2-grid cell limit, the vector is considered to be spurious and is rejected. Essentially,

this means there must be at least three consistent motion estimates adjacent to each other. These spurious vectors occur most frequently near the ice edge.

These satellite-derived motion sources have different characteristics, which influence the precision and quality of the retrieved ice motions. The different microwave frequencies and polarizations are sensitive to different aspects of the surface that may affect the cross-correlation; different frequencies also have different spatial resolutions that affect the theoretical precision (e.g., 85/91 GHz have a higher gridded resolution). AMSR-E provides substantially higher resolution that yields more precise motion estimates. AVHRR is sensitive to visible or infrared characteristics of the ice that yield a different correlation basis for feature matching. All of these differences make merging these disparate sources into a combined field inherently complex.

**Version 4 changes.** There have been two significant changes made to the satellite imagery processing for Version 4. First, the final quality-controlled and calibrated gridded SSMI and SSMIS brightness temperatures [*Maslanik and Stroeve*, 2004] have been used throughout the record. In previous versions, near-real-time gridded brightness temperatures [*Maslanik and Stroeve,* 1999] were used to augment the time series and there was no provenance on when the near-real-time or final source was used. Another change corrected over-filtering of SSMI and SSMIS vectors that removed valid motion estimates in Version 3 of the product. Motion estimates are computed using the MCC individually from SSMI and SSMIS 37 GHz and 85/91 GHz fields. In Version 3, SSMI and SSMIS vectors were only included if a similar SSMI/SSMIS vector was found in three adjacent grid cells instead of two. In Version 4, SSMI/SSMIS vectors were included if (a) there are at least two SSMI/SSMIS estimates at adjacent grid cells with similar velocities in each frequency-derived field, and (b) there are at least four similar velocities at adjacent grid cells among the combined four SSMI/SSMIS frequency-derived fields. The net effect of this change was to reduce over-filtering of valid SSMI/SSMIS-derived ice motions. This had a relatively small effect in the Arctic because the multiple motion sources provided nearby motion estimates to compensate for the lack of microwave estimates; however, in the Antarctic, where the SSMI and

SSMIS estimates provide the primary (and after 2000, the only) motion information, the sparser motion estimates often resulted in unrealistic circulation patterns; this is discussed further below in Section 2.3.

*Reanalysis winds*

The satellite imagery sources are augmented in the Arctic with motions derived from wind forcing using the NCEP/NCAR Reanalysis [*Kalnay et al.*, 2016] on a roughly 2° x 2° latitude-longitude grid, which are interpolated to a 50 km EASE-Grid (see Table 2). Wind-derived motions are not currently used in the Antarctic. The ice motions estimates are derived based on a simple relationship between winds and ice

motion. The sea ice is assumed to move in the geostrophic wind direction, as provided by the reanalysis fields, with a magnitude of 1% of the wind speed. This was implemented based on the estimate from *Thorndike and Colony* [1982]. Other studies have shown a higher percent (e.g., 2%) for the ice vs. wind speed relationship. Recent studies indicate that the ice is becoming more responsive to winds [e.g., *Spreen et al.*, 2011]. So, the 1% value used here likely underestimates the wind-driven ice speed. However, no

changes were made to the wind-derived motions for Version 4. In the supplementary material, we show that the combined motion fields are largely insensitive to the magnitude of the wind contribution because it has a relatively small weight compared to the other sources. In a future version, we plan to revisit this relationship in the Arctic and investigate adding wind-driven motions for the Antarctic.

*Buoy positions*

Ice motion vectors are also computed by incorporating position data from the network of drifting buoys deployed as part of the International Arctic Buoy Program [*IABP*, 2008]. These buoys monitor meteorological and oceanographic conditions for real-time operational requirements and research

purposes, and provide ice motion by transmitting updated locations. This product uses the twice daily (midnight and noon) locations of the IABP "C" buoy product. Two motion estimates are computed from these locations: one from noon of one day to noon of the following day, and one from midnight of a day

to midnight the following day. No buoys are included in the Antarctic motion fields because there have been few buoy deployments on ice in the Southern Ocean.

**Version 4 changes.** The principal change for the buoys is how the twice-daily observations are integrated into a daily product. Previous versions of this product considered these motions independently of each other and effectively used the most recent observation for a day. In Version 4 the two estimates are averaged to provide one daily motion estimate for each buoy. Thus, each day's buoy motion is an average of midnight to midnight (UTC) of the current day and noon the previous day to noon the current day. Also, the IABP source product recently started including floatable buoys, resulting in motion estimates from off the ice. These were not screened out in earlier versions. The effect was relatively small and primarily influenced motions near the ice-edge because of the distance-weighting interpolation. Version 4 now applies an ice mask to the buoys, making the buoy motion domain consistent with the other sources.

*Masks for valid motions*

Two masks are applied to limit motion retrievals to only regions where sea ice exists. First, a modified land mask is applied. The standard land mask is "dilated" so that cells near land are also excluded because motion retrievals near the coast are unreliable due to the effects of mixed land and ice/ocean grid cells. Because of its narrow channels, the Canadian Archipelago region is also masked out.

Second, a sea ice mask is also applied to limit motion retrievals to only ocean regions that are ice-covered on the days under consideration. The mask is based on the "Sea Ice Concentrations from Nimbus-7 SMMR and DMSP SSM/I-SSMIS Passive Microwave Data, Version 1" at NSIDC [*Cavalieri et al.*, 1996]. The mask defines all areas with concentrations greater than 15% as ice-covered so that valid ice motions can be computed.

**Version 4 changes.** Previously, the sea ice mask from only the first day was used to define the valid motion region. This was changed in Version 4 to allow motions only where ice is present on both days

used to retrieve motions. This results in very small changes near the ice edge. As noted above, the mask is now applied to buoys as well as the other sources.

## 2.2 Review of uncertainty characteristics of motion estimates from previous studies

In this section we provide an overview of general uncertainty characteristics of the source motion estimates found in previous studies, focusing particularly on passive-microwave error estimates. Errors in the ice motion and ice age products are dependent on the resolution of the satellite sensor, as well as geolocation and binning errors for each image pixel [*Meier et al.*, 2000]. The distance precision of motion detection is limited by the grid cell resolution – a pattern can nominally be "observed" to move only an

integer number of grid cells. Particularly for the low-resolution inputs, this yields high uncertainty for each individual estimate and an overall noisy motion field.

As noted above, for a 25 km gridded passive microwave input with 4X oversampling, the theoretical limit of precision of the motion is 7.23 cm/s (6.25 km/day). Atmospheric effects and temporal variability of

the surface are additional sources of error, especially in the summer. However, several evaluation studies have found that in practice errors are often lower because the different sources of error offset each other. *Kwok et al*. [1998] compared ice motion estimated from the European Space Agency (ESA) Remote Sensing Satellite (ERS-1) synthetic aperture radar (SAR) along with drifting buoy motion to SSMI-derived motions and found an error of 5-12 km/day (~6-14 cm/s). Meier et al. [2000], comparing with

buoys, found RMS errors of SSMI-derived daily velocity components to vary between ~5-7 cm/s, depending on conditions, with near-zero bias. AMSR-E, with higher spatial resolution, yields motion estimates with velocity component errors of 4-5 cm/s [*Meier and Dai*, 2006; *Kwok*, 2008].

Summertime drift error is higher due in part to surface melt, which affects the passive microwave

identification of ice parcels. Our product incorporates summer drift estimates from passive microwave, but the errors are substantially higher and the number of valid motions is lower (see supplementary material). *Kwok* [2008] showed that AMSR-E 19 GHz channels can provide improved summer estimates

compared to other frequencies. However, the large sensor footprint of 19 GHz makes such retrievals impractical except from the higher resolution AMSR-E sensor. 19 GHz was not used as an input to our product. The largest drift error was found to occur in the fall, likely due to formation of new ice [*Meier et al*., 2000]. Optimal interpolation (discussed below) reduces errors through its error and distance-based

weighting, particularly when buoys are incorporated. Temporal averaging further reduces errors in the weekly estimates.

In addition, the errors are not generally cumulative, because the motions were found to be largely unbiased evaluations done during the development of the original product; this allows for accurate tracking of

parcels (e.g., ice age) over time. These evaluations, described in the product documentation at NSIDC (https://nsidc.org/data/nsidc-0116), show *u* velocity component biases of ~±0.05 cm/s and *v*-component biases of 0.4-0.7 cm/s. Other published studies (such as the references above) show similar results. The low bias in the estimates means that errors in long-term (weeks to months) displacement are relatively small. *Tschudi et al*. [2010] compared drift tracks composed from the sea ice motion product to the drift

of the Surface Heat Budget of the Arctic Ocean (SHEBA) ice camp [*Uttal et al*., 2002] and found a drift error of 27 km over 293 days. There is some effect from the different passive microwave sources due to temporal sampling between SMMR (every other day) and SSMI/SSMIS (daily); the higher sampling rate from SSMI/SSMIS changes the discretization of the retrieved motions. Also, the higher spatial resolution of AMSR-E affects the discretization of the motions as well. This is discussed further in the

supplementary material.

We note here that evaluation of sea ice motions has come nearly exclusively from the Arctic region. The primary reason for this is the existence of the IABP buoys that offers a reliable "truth" for evaluation of satellite-derived motions and other methods. The Antarctic has had few or no buoys. Thus, our knowledge

of the error characteristics of Antarctic motions have not been quantified. While the cross-correlation approach for the satellite-derived motions is the same in the Antarctic and the theoretical precision is thus the same, the Antarctic sea ice surface is different (e.g., thinner ice, deeper snow, snow-ice formation). Other factors, such as a more dynamic sea ice cover and different atmospheric influence also have an

effect. As such, there is lower confidence in Antarctic motions and the error characteristics are more uncertain. The Antarctic motions are included with the product for completeness, but users should note these caveats.

## 2.3 Combined gridded sea ice motion fields

Daily motion fields are provided from each of the sources during their period of availability. However, for many users, the most useful parameter is the combined gridded product. This combines via an optimal interpolation scheme all available sources for a given day onto a version of the 25 km EASE-Grid [*Brodzik et al.*, 2002]. For further information on the grid, see NSIDC's documentation for this data product [*Tschudi et* al., 2019a]. For each 25-km ice EASE-grid cell, the speeds (cm/s) in the EASE-grid x-direction (*u* velocity component) and y-direction (*v* velocity component) are stored. The daily motions fields are also averaged into weekly fields.

Optimal interpolation (also called "kriging") is not simply a spatial average, but considers the accuracy of different sources and the spatial distribution of the source estimates. The motion estimates vary in expected quality, with buoys considered most accurate, followed by passive-microwave and/or AVHRR-based estimates and finally by the wind field. This weighting is of the form:

$$w = Ce^{(-d/D)} \tag{1}$$

where $w$ is the weight, $C$ is a source-based coefficient (0.45 for wind, 0.95 for buoy, 0.8 for other sources), $d$ is the Euclidean distance between the pixel in question and the motion estimate on the EASE grid, and $D$ is the length-scale (constant) over which the estimates are correlated. The values of $C$ are constant for each source and are based on early comparisons between each source and buoy estimates. Buoys, being the most accurate, were assigned the 0.95 value. The buoys were used as the baseline for estimating the other weights. The values of $C$ for the other sources were estimated *a priori* based on comparisons between the source motions and buoy estimates. The original derivation of the $C$ values was not retained; it is likely that the values are not optimal in all cases. For example, the quality of the satellite estimates

varies depending on source and spatial resolution, so using 0.8 for all of them is sub-optimal (see supplementary material). Another example is that wind-derived estimates appear to be comparable to many of the satellite estimates (see supplement), suggesting that winds should be weighted relatively higher. However, these were not changed for Version 4 and here we simply provide the values used in the product.

Estimates that are closer (low $d$) and higher quality (sources with higher $C$, e.g. buoys), are weighted higher. The correlation length-scale, $D$, was given a value of 417 km, also determined empirically, based on cross-correlations of estimates separated by varying distances. This distance is lower than the full correlation length scale. However, the method limits the number of interpolated source observations to a maximum of 15 and this distance is large enough to encompass that limit. Other studies (e.g., Meier et al., 2000) found that using longer length scales did not appreciably affect the interpolation values. The method loops through all grid cells in the domain that are flagged as sea ice-covered. Figure 2 shows an example of the individual motion sources and the resulting combined motion field. The optimal interpolation converts the sparse and/or noisy individual motion fields into a complete and smoothly varying combined motion grid.

**Version 4 changes.** The most notable change in the motion product for Version 4 involves the optimal interpolation approach. In previous versions, the combined estimate at each valid grid cell was estimated by optimally interpolating (kriging) the surrounding 15 closest vectors. While this generally gives a good spatial distribution around grid cells, it does not necessarily include all estimates that fall within correlation length-scale and that theoretically could influence the interpolated estimate. This means that discontinuities can potentially occur, particularly as highly-weighted estimates (i.e., buoys) fall off the list of closest estimates. When the buoy motion estimates differ significantly from other sources, artificially large spatial gradients in velocity magnitude can arise [*Szani et al*., 2016]. In Version 4 of the product, the methodology has been revised to use the 15 highest-weighted ice motion vectors at each grid cell, regardless of source. So, a source with a high value of $C$ (i.e., buoys) will have a weight, $w$, higher than $w$ for a source with a lower $C$ value (e.g., winds) over a longer spatial distance.  This means that

higher weighted observations have influence over a longer distance and their influence drops off more gradually. This approach significantly reduces and often removes the discontinuity artifact in the daily combined product (Figure 3). It is also reflected in the interpolation error estimates included with the daily product (not shown), where the low error in the neighborhood of the buoys has a smoother gradient.

As noted above, the Version 4 algorithm also eliminates an over-filtering of SSMI and SSMIS passive microwave vectors that occurred in Version 3. Since these vectors are the primary source in the Antarctic (other than SMMR during 1978-1987 and AVHRR during 1981-2000), they are the main input to the optimal interpolation, and the over-filtering of the vectors resulted in a sparse raw motion field. With the

length scale value, $D$, of 417 km, given the coarse spatial resolution and high noise in the daily passive microwave derived motion vectors, such few vectors often did not provide a spatially representative sample of the large-scale motion circulation. In other words, the interpolated motion field of the Antarctic ice motion field was often being driven by very few underlying motion estimates, which led to unrealistic circulation patterns in the Antarctic because there were too few vectors to create a representative field.

The over-filtering also occurred in the Arctic but was much more limited because other sources exist to augment the passive microwave estimates; in particular, use of spatially complete and smoothly varying wind-driven motions in the Arctic "filled in" any place where passive microwave vectors were sparse (see Figure 2). Version 4 corrects this over-filtering, yielding more passive microwave motion estimates over a broader area; this is particularly noticeable in the Antarctic. An example of this is shown in Figure 4

where the Version 3 product has very few vectors. In the eastern Weddell Sea, this results in southward onshore ice motion. This would be very unusual for the region and comparisons with winds (not shown) indicate that this motion is not realistic. Version 4 yields more source vectors that better represents the spatial variation in the region. The result is a general eastward circulation, which is more typical for the region and is consistent with the wind field.

A final change in the motion product for Version 4 is that the data are now provided in NetCDF format, with daily files for each underlying motion field, e.g. SSMI, buoy and wind-driven motions – as well as files containing the daily combined (optimally interpolated) estimate and a weekly average sea ice motion.

The self-describing file format provides improved metadata (including georeference information) and easier access for many users. In the daily combined field, an error estimate is included that gives the error from the optimal interpolation, which is a function of the number, spatial distribution, and quality of all input vectors interpolated at a given grid point. Flag values are used to denote potential low-quality interpolation due to lack of nearby vectors and/or vectors near the coast (where retrievals have higher errors). Because the passive microwave daily ice motions are at a coarse resolution, they tend to exhibit discretization effects at daily timescales (e.g., Lavergne et al., 2010), even when applying the 4X oversampling. These effects are diminished in the weekly fields as day-to-day "noise" in the observations are averaged out over the seven days. Thus, the weekly sea ice motion fields are the recommended product for most applications; users of the daily product should recognize its limitations and use caution in interpreting features and changes in the daily fields. The NSIDC archive also provides browse imagery of the weekly sea ice motions (Figure 5), which has also been updated to improve visual appearance.

## 3 The EASE-Grid Sea Ice Age Product

The EASE-Grid Sea Ice Age product [*Tschudi et* al., 2019b] builds upon the combined motion product and is also a popular dataset with over 650 unique users having accessed the data as of this writing [*NSIDC*, personal comm.]. Version 2 of the sea ice age data is also part of NASA's Making Earth System Data Records for Use in Research (MEaSUREs) dataset at NSIDC [*Anderson et al*., 2014]; however, the MEaSUREs product is not regularly updated and does not include the newest enhancements described here. Animations of motion and age have been posted on NOAA's ClimateWatch online magazine (http://www.climate.gov/news-features/videos/old-ice-arctic-vanishingly-rare), as well as the NASA Scientific Visualization Studio (https://svs.gsfc.nasa.gov/4750). Sea ice age distributions and trends are described annually in the Arctic Report Card [*Perovich et al.*, 2019] and have been analyzed by *Maslanik et al.* [2007; 2011].

The Sea Ice Age product was introduced by *Fowler et al.* [2004] and described further by *Maslanik et al.* [2007; 2011], *Tschudi et al.* [2010], and *Stroeve et al.* [2011]. The ice age product algorithm estimates the age (in years) of Arctic sea ice using input from the previously described sea ice motion product.

Weekly averaged motions are used to reduce computational complexity and to temporally average discretization artifacts in the daily motion data. Also, the 25 km resolution motions are bilinearly interpolated to a 12.5 km resolution grid in order to provide finer granularity in the ice age fields.

5   At the beginning of the ice motion record, all parcels in the 12.5 km ice age grid are initialized with an age-class of "first-year ice", meaning ice that is less than one year old. These parcels are then treated as Lagrangian particles and are advected at weekly time steps with the motion product estimates. When two or more parcels merge into a grid cell, the age of that grid cell is represented as the age of the oldest parcel. Rarely, ice motion results in all parcels being advected out of a grid cell; when this occurs, a new
10  parcel of "first-year ice" is initialized in that grid cell. During the week of the Arctic sea ice extent minimum, the age of all parcels is increased by one year. At each time step, all parcels found within a grid cell that have an ice concentration of less than 15% are considered to have melted and are no longer considered in determining the ice age. Parcels are tracked for up to 16 years, after which they are no longer considered (such parcels are simply removed).

This approach does not consider new ice that may form within a grid cell because it retains only the oldest ice in its accounting. Thus, the product is effectively an estimate of the oldest ice in a given grid cell. Tracking of partial concentration of age categories can provide a more detailed picture of the ice cover [*Korosov et al.*, 2018] and is something we may consider for future versions.

The source motion data for the age product begin in 1978, with the age of all parcels initialized as first-year (0-1 years old) ice. Because the method tracks age over time, several years are needed to build up older ice categories. For this reason, the ice age product begins in 1984. The youngest ice age category is first-year ice (FYI), which is ice that is less than a year old. Similarly, second-year ice is one to two years
25  old, and so on for older ice age categories. Ice older than 4 years (5th-year and older ice) makes up a very small percentage of the ice cover, so depicting ice older than this category as a separate field in browse imagery is not undertaken. Therefore, the ice age is frequently categorized as being of ages: 0-1 (i.e., FYI), 1-2, 2-3, 3-4, and more than 4 years old (i.e., 5th-year and older ice).

**Version 4 changes.** The primary changes in Version 4 of the ice age product result from the changes in the source ice motion products described above. The most substantial change addressed anomalous behavior in the motion and age fields documented by *Szanyi et al.* [2016]. They showed that

discontinuities in the interpolated motion field, caused by sub-optimal interpolation of buoys with the other data sources, created artificial ice divergence and new ice formation in the Version 3 product. This potentially results in an underestimation of multi-year and an overestimation of first-year ice. The change in the interpolation weighting, described above, reduced this effect as seen in Figure 3; the Version 3 field has the circular features surrounding the buoys where the buoy contribution suddenly drops out, resulting

in a discontinuity where false divergence can occur. The new weighting scheme smooths that discontinuity and eliminates much of the false divergence. The effect of this change can be seen qualitatively in the age fields as less "speckling" of first-year ice interspersed within the multi-year ice pack; the age fields show a more realistic consolidated multi-year ice pack. Qualitatively, the net effect is less first-year ice (the "speckling" that results from the false divergence) and an increased amount of

multi-year in Version 4 compared to Version 3 (Figure 6). This effect becomes much less noticeable during the latter part of the record. There are three reasons for this. First, there is less passive microwave coverage during the early SMMR period, so a sparser number of vectors, which will accentuate interpolation-induced artifacts in the data. Similarly, in the early part of the record, there were far fewer buoys, so the buoy interpolation discontinuities are more noticeable. In recent years, there are enough

buoys such that the interpolation distances of neighboring buoys often overlap, so discontinuities with the passive microwave and wind fields are less common. Finally, there is simply much less multi-year ice in recent years, so the discontinuity effects are less pronounced. A quantitative assessment of the version changes in the ice age product are discussed further in Section 4 below.

Two other minor changes to the ice age product have been introduced in Version 4. First, the week-numbering convention was slightly modified to be consistent with the motion weeks. Second, browse imagery (Figure 6) was improved to explicitly show ice-covered ocean areas that are outside of the age and motion domain (e.g., the Canadian Archipelago).

Validation of sea ice age is difficult because there is no known suitable validation data set that can be used for a comparison. Here we primarily rely on the fact that the ice age product is directly derived from the ice motion product. Thus, the demonstrated improvement in the motion fields indicates that the age fields are also improved. This is particularly noticeable in the reduction of the circular features in the motion field, which reduces the "speckling" in the Version 4 age fields. While this is qualitative, we feel this does demonstrate an improvement in the age fields. A recent study [*Lee et al.*, 2017] included the NSIDC ice age fields in a comparison with passive microwave ice age retrieval methods, including multiyear fraction from the NASA Team algorithm, the OSI-SAF ice type product [*Aaboe et al.*, 2017], and a microwave emissivity approach. The spatial patterns of first-year and multi-year ice in the NSIDC age product matched well with the comparison products, showing that our age product is at least consistent with other approaches.

## 4 Trends and variability in Version 4 ice motion and age and comparison to Version 3

Here we evaluate how the changes from Version 3 to Version 4 of the products affect the long-term trends and variability in the sea ice age fields. We also provide updated motion and age trends through 2017.

As seen in Figure 3, the change to Version 4 does noticeably affect parts of the daily fields in regions around buoys. Over a weekly period, the changes are less significant because the temporal averaging smooths out the variability in the motions. The weekly average speed is generally faster in Version 4 than in Version 3 (Figure 7). (The differences in the $u$ and $v$ motion components (not shown) have similar characteristics over the time series.) This change in speed between Version 3 and Version 4 reflects the two major changes made for Version 4: (1) the use of the 15 highest weighted observations for the interpolated combined fields, and (2) the correction of the over-filtering of the SSMI and SSMIS vectors. During the SMMR part of the record, the differences are generally near-zero. This is because only the change in weighting had an effect on this period. In the Arctic this primarily changed the influence of the buoys and there were fewer buoys during the SMMR period. In the Antarctic (Figure 7b), the change is even smaller because there are no buoys and thus less impact of the adjusted weighting scheme. For the

SSMI and SSMIS period, the Version 4 motions are ~0.5 – 1.0 cm/s faster than Version 3 in the Arctic and ~0 – 2 cm/s faster in the Antarctic. In this period, both the change in weighting and the over-filtering correction affected the motions. The over-filtering effect on the number of valid SSMI and SSMIS has larger effect, especially in the Antarctic where there are no buoys or wind-derived fields. During 2002-

5   2011 when AMSR-E is included, the Arctic speed difference is reduced with Version 4 speeds ~0.25 – 0.5 cm/s faster. AMSR-E motions did not change between Version 3 and 4; SSMI and SSMIS were also used in this period, but with higher resolution, more AMSR-E motions were used. Thus, the over-filtering issue in the SSMI-SSMIS estimates was muted in the AMSR-E period. After the end of AMSR-E the differences increase again. In the Antarctic, there is no notable change because AMSR-E is not used.

There is seasonal variation with larger differences during the Arctic summer. The main factor is likely overall speeds, as seen in the Version 4 weekly average speed timeseries (Figure 8) that show strong seasonal variability in Arctic motions with speeds peaking during summer. In the Antarctic, the version differences are actually largest in winter and smaller in the summer; this may reflect fewer vectors with

15   minimal summer ice cover. Also, in the Antarctic winter, the ice extends far northward and the pack is quite dynamic in response to winds and currents. Other factors also play a role, including the number of vectors from different sources at different times of year (e.g., fewer passive microwave vectors during summer) and, in the Arctic, the revised weighting scheme that effectively yields more influence of buoys during the summer (when there are fewer passive microwave motions).

There is also interannual variability (Figure 8), some of which is related to the SMMR every-other-day sampling, resulting in slower speeds and less variability for the 1979 to 1987 period (more noticeable in the Antarctic because of the lack of wind-derived and buoy motions). Beyond that, there is an overall positive trend in Arctic sea ice speed of 0.21 cm/s/decade in Version 4 versus 0.13 cm/s/decade in Version

25   3. The increasing speed is in general agreement with previous studies that noted a trend toward faster moving ice [e.g., *Spreen et al.*, 2011] and linked the trend to greater response to wind-forcing by a thinner ice cover. In the Antarctic, there is also an increasing trend of 0.61 cm/s/decade in Version 4 versus 0.41 cm/s/decade in Version 3. But as noted above, the differences in the trend values from Version 4 and

Version 3 at least partially reflect the effects of the changes in the motion sources and their relative impacts over the time series.

The largest effect of the version change for ice age is, as noted above, the amount of multi-year ice in the early part of the record, particularly in the oldest ice categories. This is illustrated in the timeseries of ice age (Figure 9). Both versions show a strong decline in 4+ year old ice over the record, with a steep loss of old ice in the late 1980s through the mid-1990s, which is associated with a persistent positive mode of the Arctic Oscillation (AO) [*Rigor et al.*, 2002]. A positive AO results in increased drift from the Siberian coast and greater advection of ice out of the Arctic through Fram Strait, which serves to "drain" older ice out of the Arctic [*Rigor and Wallace*, 2004].

The change to Version 4 results in higher extent of the old ice over most of the early part of the record, with the exception of 1995-1996 (perhaps related to the end of the positive AO period and/or large changes in minimum extent between the two summers). Version 4 extent of 5+ year old ice is on average 367,000 km$^2$ higher than Version 3 for the first five years (1984-1988) of the record. This is an effect of the improved interpolation weighting scheme and is a quantitative indication of the reduced "speckling" discussed earlier. However, the impact dissipates over time; during the last five full years of the record (2012-2016), the difference between Version 4 and Version 3 is only 42,000 km$^2$. The amelioration of the difference is likely due to two factors: (1) the transition from SMMR to SSMI-SSMIS and the resulting improved coverage; and (2) the increasing number of buoys over time. As noted above, the two-day SMMR separation does change the motion discretization and the spatial coverage. So, the relative effect of the buoys is greater during the SMMR era. And as buoy coverage increases over the years, there is more overlap in buoy influence, so the change in weighting that increases the distance of buoy influence has a relatively smaller effect. With daily data and better spatial coverage in SSMI, the differences between the two versions starts to decrease. This decrease continues as buoy coverage increases over the years. And with AMSR-E and its better spatial resolution added in 2002, the differences drop further as the AMSR-E motions start affecting the older ice types in the following years. By 2005, there is very little difference between the two versions. Focusing on the week of 19-25 February, the larger differences

between versions of 4+ year old ice compared to younger ice types is evident (Figure 10). The younger ice categories show smaller, generally negative differences (i.e., less younger ice in Version 4). Thus, the changes in Version 4 appear to improve the ice age fields by removing much of the artificial divergence noted in *Szanyi et al.* [2016], thereby reducing the amount of younger ice and increasing the amount of older ice. However, the impact of the version change decreases over time such that there little impact on the age distribution in recent years.

Both versions of the ice age field show a transition from one dominated by older ice to one dominated by younger ice (Figure 11). Interannual variability is evident in all ice age classes, particularly first-year ice, which is not surprising given the variability of the summer ice cover. Less variability is seen in older ice. Nonetheless, the decline in older ice is apparent during the late-1980s through the mid-1990s persistent positive mode of the Arctic Oscillation [*Rigor et al.*, 2002]. After 1994, there was some recovery in multi-year ice before beginning a significant decline after 2004. Linear trends are estimated for the Arctic Ocean region. This is a region bounded by the northern coasts of the continents, the Bering Strait, Fram Strait, and the ~20 E meridian between Svalbard and the Fennoscandian Peninsula. The total area of the region is ~$7.8 \times 10^6$ km$^2$. Using this region excises areas where only first-year ice exists, so it focuses on the areas where there is variability in the ice age. There is a strong increasing trend in ice less than 1 year old (Table 3) and a similar decreasing trend in 4+ year old ice. Trends in the intermediate ages (1-4 years old) are smaller. This is partly due to smaller extents of these ages as well as the fact that ice transitions through these categories between the larger extents of the oldest and youngest ice.

## 5 Conclusions

New versions (4.0) of the sea ice motion [*Tschudi et al.,* 2019a] and sea ice age [*Tschudi et al.,* 2019b] datasets have been produced and are now available at NSIDC. Routine updates will regularly occur when the underlying data – buoy positions, brightness temperature fields and sea ice concentration fields – become available. This is expected to occur every few months.

Arctic sea ice motion vectors are currently constructed by merging motion vectors estimated using three sources: buoys, passive microwave satellite imagery, and winds from NCEP/NCAR. In the Antarctic, only the satellite imagery vectors are used. Sea ice age is produced for the Arctic using the weekly sea ice motion product as input, tracking ice parcels and aging them each year if they neither melt nor advect

out of the ice pack.

The most recent sea ice motion algorithm revision incorporates improvements such as an improved vector weighting scheme, corrections to passive microwave vectors, new browse imagery, and the underlying code base through the use of Python. Furthermore, the Version 4.0 upgrade addresses artifacts in the ice

motion resulting from the interpolation. These artifacts did have a noticeable effect on the weekly motion and age fields early in the record, but in more recent years, the effect of these artifacts is diminished due in large part to many more buoys in the Arctic, which results in overlapping influence of buoys and thus fewer artifacts.

We note the decrease in older sea ice over the ice age record, from the 1980's, when older ice constituted ~30% of the ice pack, to recent years, when older ice occupies less than 5% of the pack. *Tschudi et al.* [2016] compared ice age to ice thickness derived from ICESat [*Kwok et* al., 2009; *Kwok and Cunningham*, 2008] and NASA's IceBridge campaign [*Kurtz et al.*, 2012, 2013]. They found that the thickness/age relationship has an approximate linear fit for the ICESat dataset, but that the relationship was much more

variable for IceBridge, due to the Arctic basin-wide coverage of ICESat thickness data and the more limited areal coverage for IceBridge aircraft-acquired data. The relationship found between ice age and thickness for the basin-wide ICESat dataset suggests that the ice age product may be used as a general indication of the sea ice thickness distribution, and could be compared to other Arctic basin-wide sea ice thickness estimations, such as those from CryoSat-2 [*Salilla et al.,* 2019].

The ice motion and age products are continuously being improved. We plan to utilize passive microwave imagery from the AMSR2 instrument aboard the GCOM-1 satellite in a future release of the motion product, which may reduce the error in motion, due to the improved higher spatial resolution of AMSR2

over SSMIS. We also plan to further improve the age product by categorizing the age distribution in each EASE grid cell (as suggested by *Korosov et al.*, [2018]), instead of retaining only the oldest ice age. Other improvements in the sea ice motion and age products are under consideration.

## 6 Acknowledgments

The authors thank the anonymous reviewers for their helpful reviews of this manuscript. Research for this manuscript is supported by the NASA Cryospheric Sciences Program, per award NNX16AQ41G, and the NASA Snow and Ice Distributed Active Archive Center (DAAC) at NSIDC.

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

**Table 1.** Version histories of the sea ice motion and age products.

| Version | NSIDC Release Date | Motion | Age |
|---|---|---|---|
| 1 | Not distributed by NSIDC | Original version based on SMMR, SSMI, and AVHRR imagery, and buoy motions | Original research product |
| 2 | Sep 2013 (motion) Dec 2014 (age) | • Added AMSR-E sources<br>• Added NCEP/NCAR wind-derived motions for Arctic | • First version distributed at NSIDC (as Version 2)<br>• Used Version 2 ice motion product as input |
| 3 | Feb 2016 | • Removed erroneous buoy and AVHRR-derived motions<br>• Updated buoys motions through most recent date<br>• Derived sea ice mask from NSIDC* product instead of internally derived concentration estimates<br>• Used GDAL** library to interpolate SSMI fields from polar stereographic to EASE grid<br>• Improved browse images | • Used Version 3 ice motion as input<br>• Improved browse images |
| 4 | Nov 2018 | • Used highest-weighted vectors for interpolated gridded fields instead of nearest vectors<br>• Daily buoy motions averaged instead of using latest observation<br>• Open water buoys removed<br>• Final quality-controlled SSMI and SSMIS brightness temperatures used throughout record<br>• Corrected over-filtering of SSMI and SSMIS vectors that had removed valid motion<br>• Improved browse images<br>• Removed monthly average fields from the product. | • Used Version 4 ice motion input<br>• Updated week-numbering convention to be consistent with motions<br>• Improved browse images |

*Cavalieri et al., 1996; **Geospatial Data Abstraction Library (https://gdal.org).

**Table 2.** Temporal coverage of input source data, as of Nov. 2019. The products will be updated approximately yearly. Buoy motions are from GPS location data. *NCEP-NCAR winds are on T62 Gaussian grid, which ~100 km in the latitudinal direction, with variable longitudinal spacing.

| Data | Source | Temporal Range | Source Resolution (km) | Gridded Motion Resolution (km) |
|------|--------|----------------|------------------------|--------------------------------|
| Daily Sea Ice Motions | Interpolated from input data | 01 Nov1978 – 31 Dec 2018 | | 25 |
| Weekly Sea Ice Motions | Averaged from Daily Sea Ice Motions | 05 Nov1978 – 31 Dec 2018 | | 25 |
| Input Data | AMSR-E | 19 Jun 2002 – 08 Aug 2011 | 6.25, 12.5 | 37.5 |
| | AVHRR | 24 Jul 1981 – 31 Dec 2000 | 5 | 50 |
| | IABP Buoys | 18 Jan 1979 – 31 Dec 2018 | NA | NA |
| | NCEP/NCAR U-wind and V-wind | 25 Oct 1978 – 31 Dec 2018 | ~100* | 50 |
| | SMMR | 25 Oct 1978 – 08 Jul 1987 | 25 | 75 |
| | SSM/I | 09 Jul 1987 – 31 Dec 2006 | 12.5, 25 | 75 |
| | SSMIS | 01 Jan 2007 – 31 Dec 2018 | 12.5, 25 | 75 |

**Table S2.** Validation statistics from comparison with CRREL buoys.

| | u-component (cm/s) | v-component (cm/s) |
|--|--------------------|--------------------|
| **Bias** | | |
| Version 3 | -0.115 | -0.687 |
| Version 4 | -0.111 | -0.660 |
| **Error St. Dev.** | | |
| Version 3 | 4.20 | 4.32 |
| Version 4 | 3.90 | 4.03 |

10  **Table S3.** Comparison with CRREL buoys of combined motions using different wind-speed scaling.

| | u-component (cm/s) | v-component (cm/s) |
|--|--------------------|--------------------|
| **Bias** | | |
| 1% of wind speed | -0.267 | -0.190 |
| 2% of wind speed | -0.263 | -0.220 |
| **Error St. Dev.** | | |
| 1% of wind speed | 4.95 | 4.43 |
| 2% of wind speed | 4.47 | 3.99 |

**Table 3.** Linear trends for ice ages over three periods. The main values are for Version 4, with Version 3 values in italics on the line below. These values are for the Arctic Ocean region.

| Sea ice age | 1984-2017 Trend [km²/year] | 1984-1996 Trend [km²/year] | 1997-2017 Trend [km²/year] |
|---|---|---|---|
| 0-1 | 69,200 | 96,200 | 92,500 |
| | *(67,700)* | *(94,000)* | *(95,600)* |
| 1-2 | 10,500 | 22,100 | 4,500 |
| | *(4,900)* | *(18,000)* | *(-3,500)* |
| 2-3 | -4,900 | 10,000 | -12,500 |
| | *(-7,300)* | *(2,000)* | *(-11,900)* |
| 3-4 | -10,100 | -11,400 | -16,100 |
| | *(-11,200)* | *(-9,000)* | *(-16,700)* |
| 4+ | -75,500 | -104,100 | -93,300 |
| | *(-64,800)* | *(-91,000)* | *(-88,000)* |

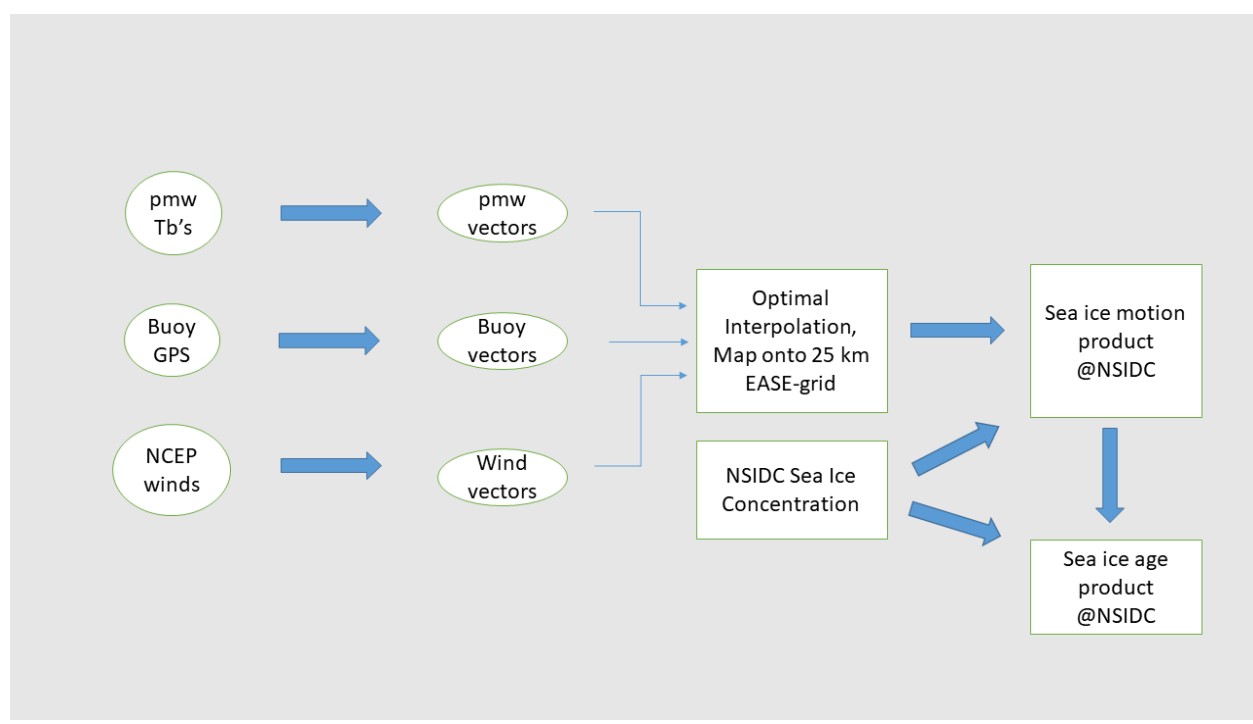

**Figure 1.** Flow chart for the production of the sea ice motion and age products.

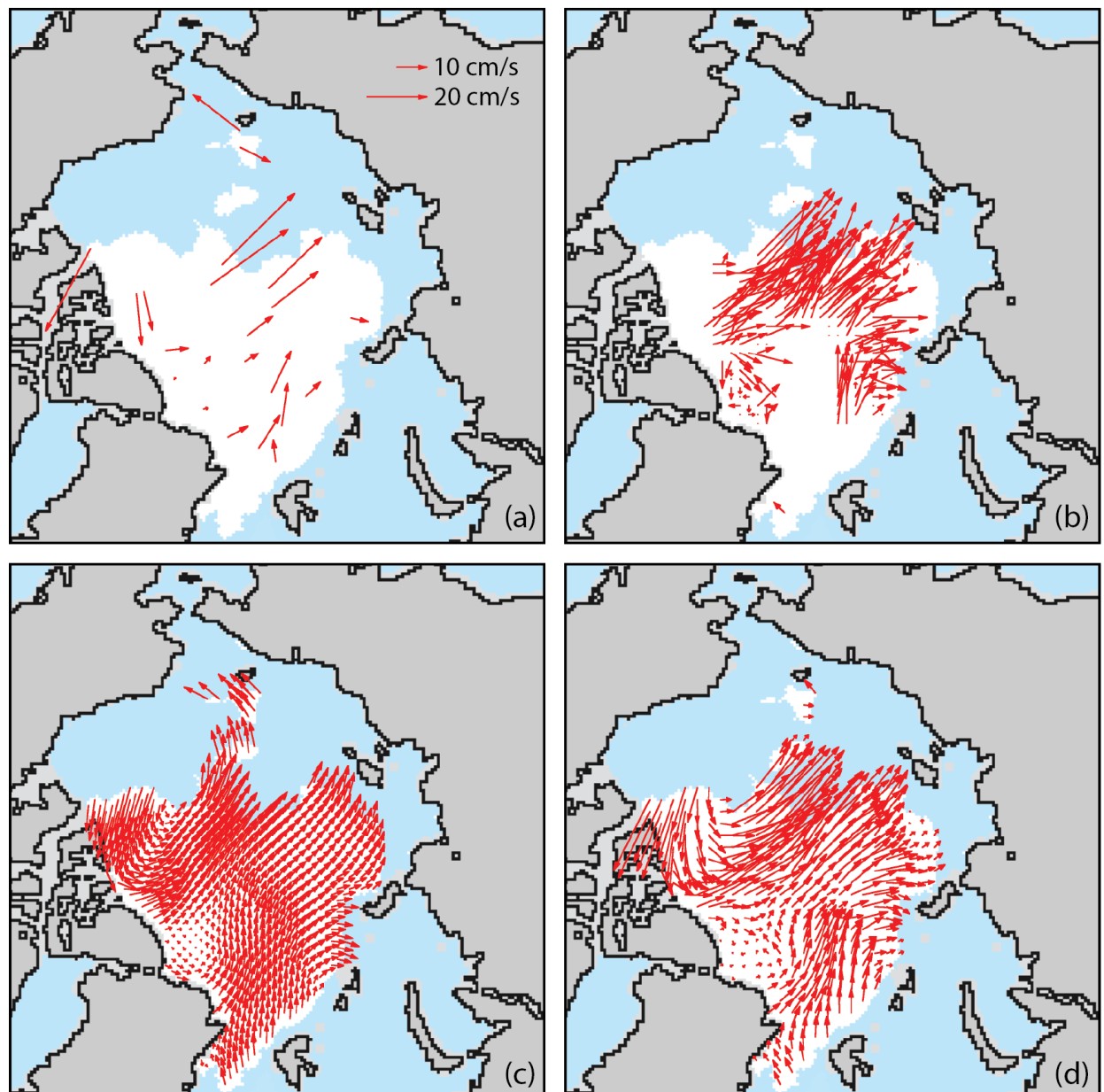

**Figure 2.** September 16, 2016 daily motion vectors from (a) buoys, (b) passive microwave, and (c) winds. The three sources are then merged to form (d) the daily interpolated sea ice motion field. Sea ice (white), ice-free ocean (blue), land (gray) and coast (black) are also shown. All buoys are shown, but other fields show only every 4th vector for legibility. In some years, AMSR-E or AVHRR also contribute vectors.

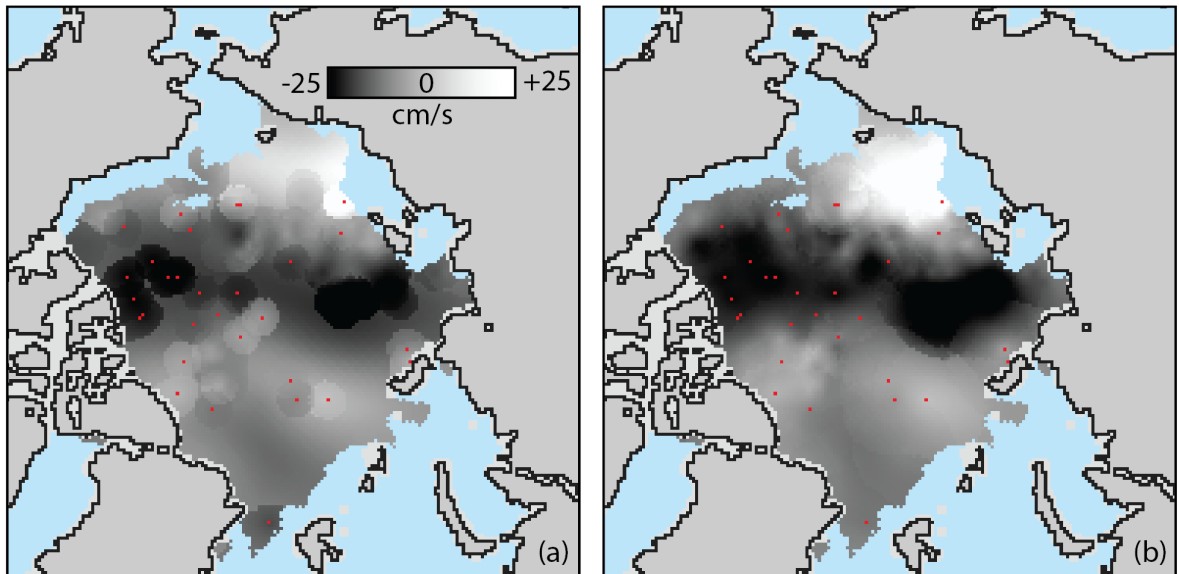

**Figure 3.** U-component of the daily interpolated vector field for September 17, 2001 from (a) Version 3 and (b) Version 4. The Version 3 fields show sharp gradients in the velocity when highly-weighted buoy estimates – buoy locations shown with red dots – no longer contribute to the motion field. Version 4 removes these sharp gradients by considering the highest weighted - rather than closest - underlying estimates.

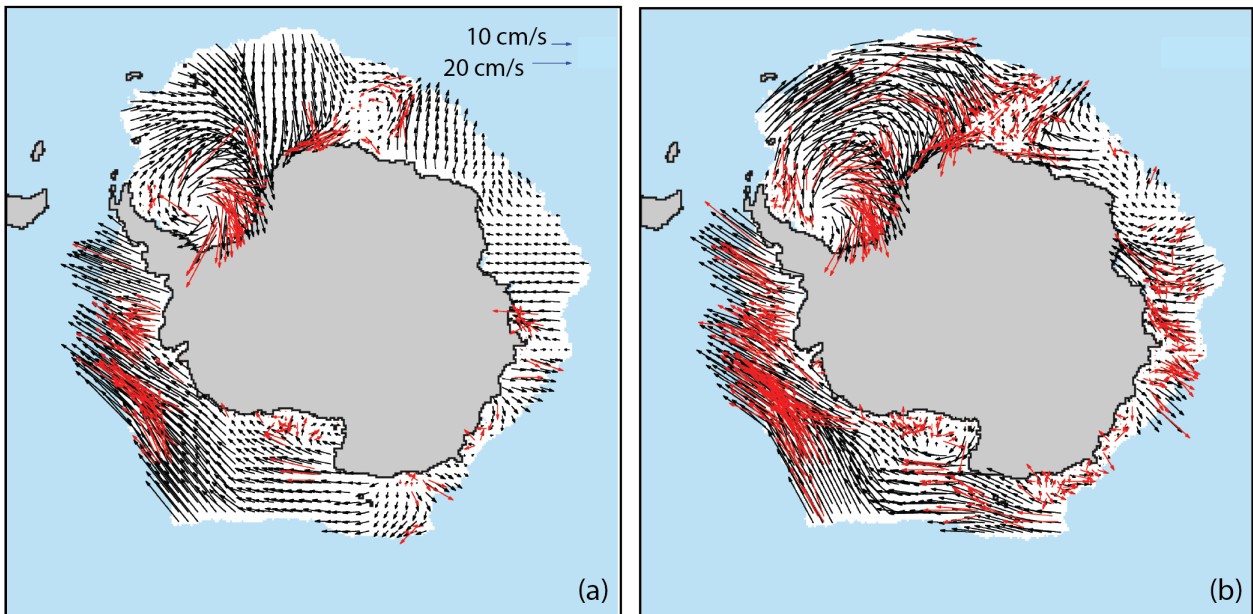

**Figure 4.** Version 3 (a) and Version 4 (b) Antarctic SSMI vectors (red) and resulting interpolated vectors (black) for August 22, 2001. Version 3 over-filtered the number of underlying SSMI vectors, often resulting in an ice field constructed from very sparse underlying data. Version 4 corrected this and includes more SSMI vectors. Every 4[th] vector is plotted for easier legibility.

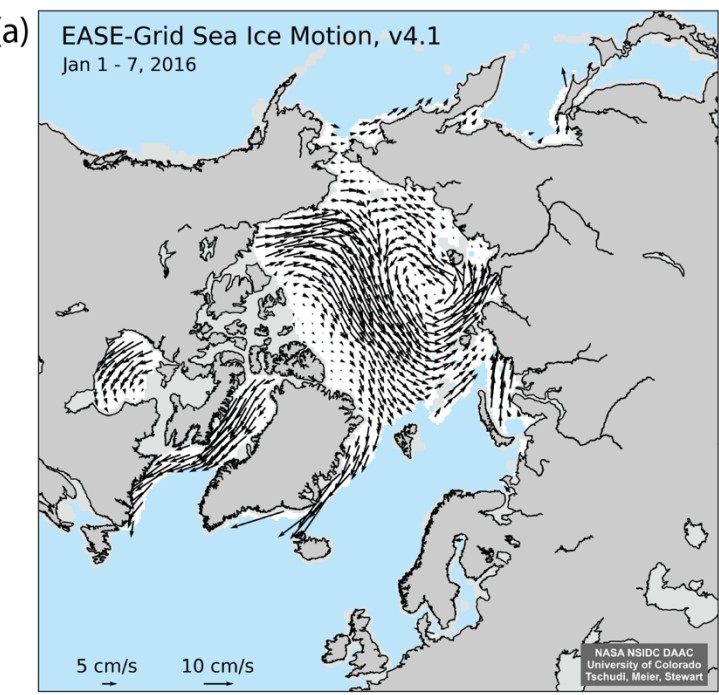

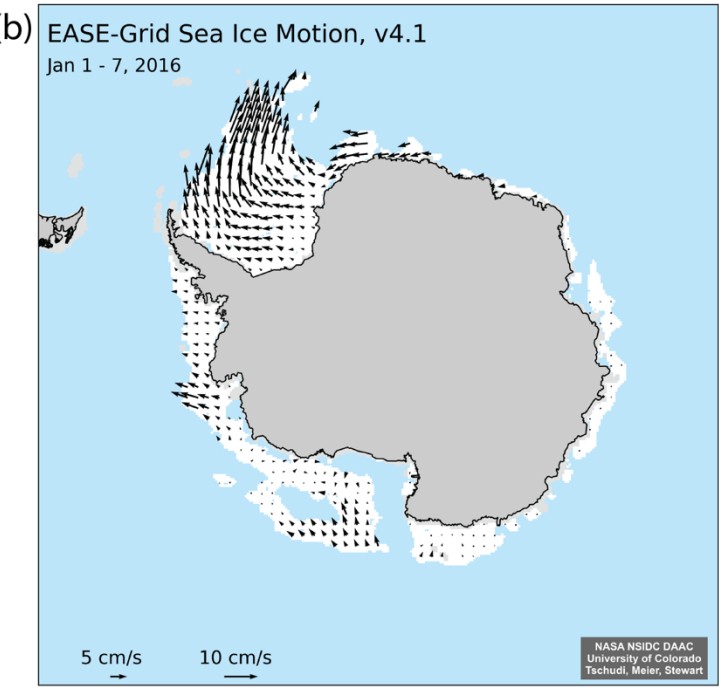

**Figure 5.** Example EASE-Grid sea ice motion for the Arctic region, the week of January 1-7, 2016 for (a) Arctic, and (b) Antarctic. White indicates the sea ice mask region (>15% concentration). Note that motions are not retrieved in the Canadian Archipelago region or near coasts in the Arctic. Every 4ᵗʰ vector is plotted for easier legibility.

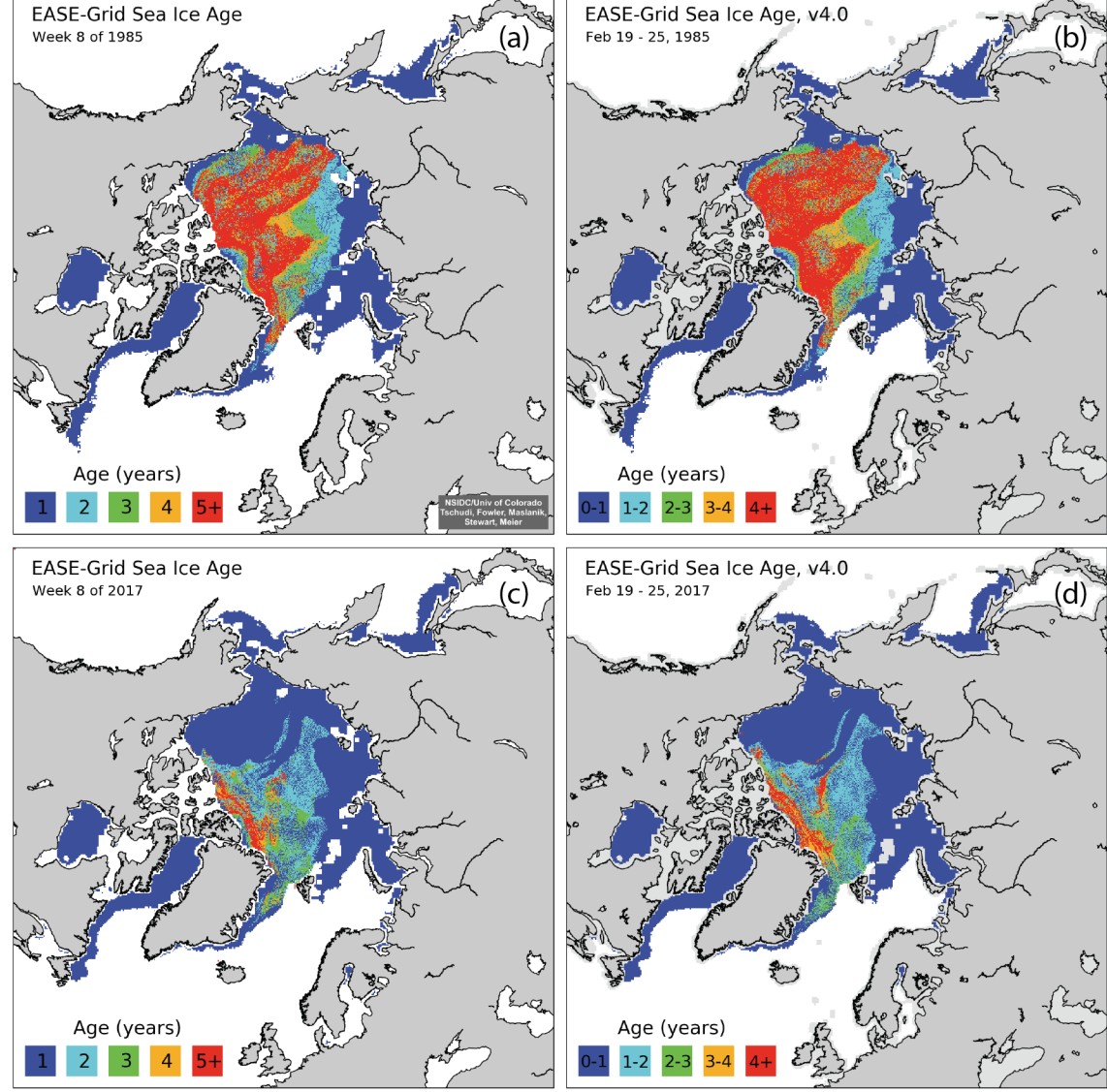

**Figure 6**. Comparison of Week 8 (Feb 19-25) ice ages for 1985 (a) Version 3 and (b) Version 4, and for 2017 (c) Version 3 and (d) Version 4.

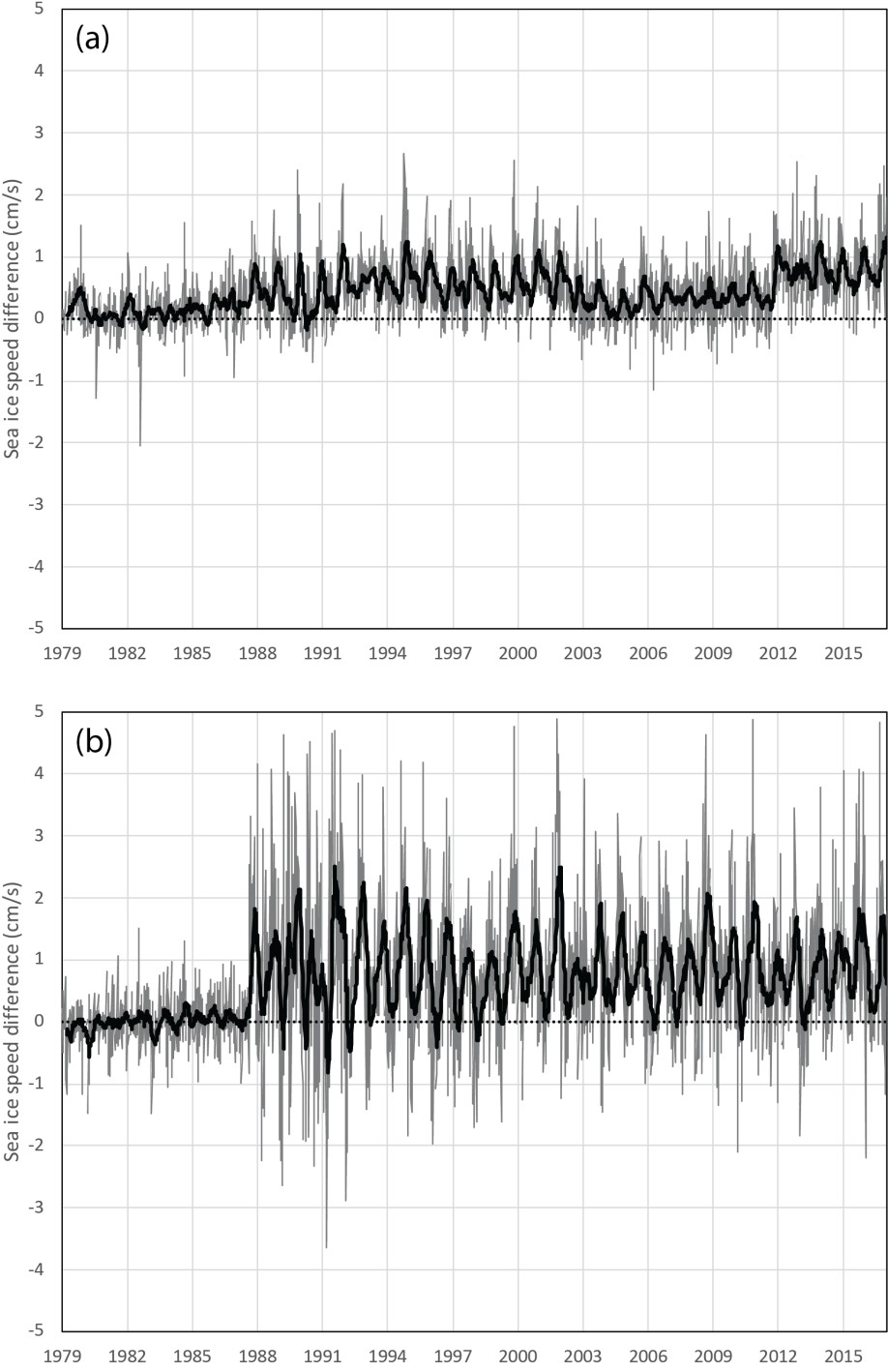

**Figure 7.** Arctic weekly average sea ice drift speed difference between Version 4 and Version 3 (V4-V3), 1979-2017. A 13-week running average is overlaid on the weekly values to highlight seasonal variability. The weekly average value is derived by averaging all vectors in the weekly motion field.

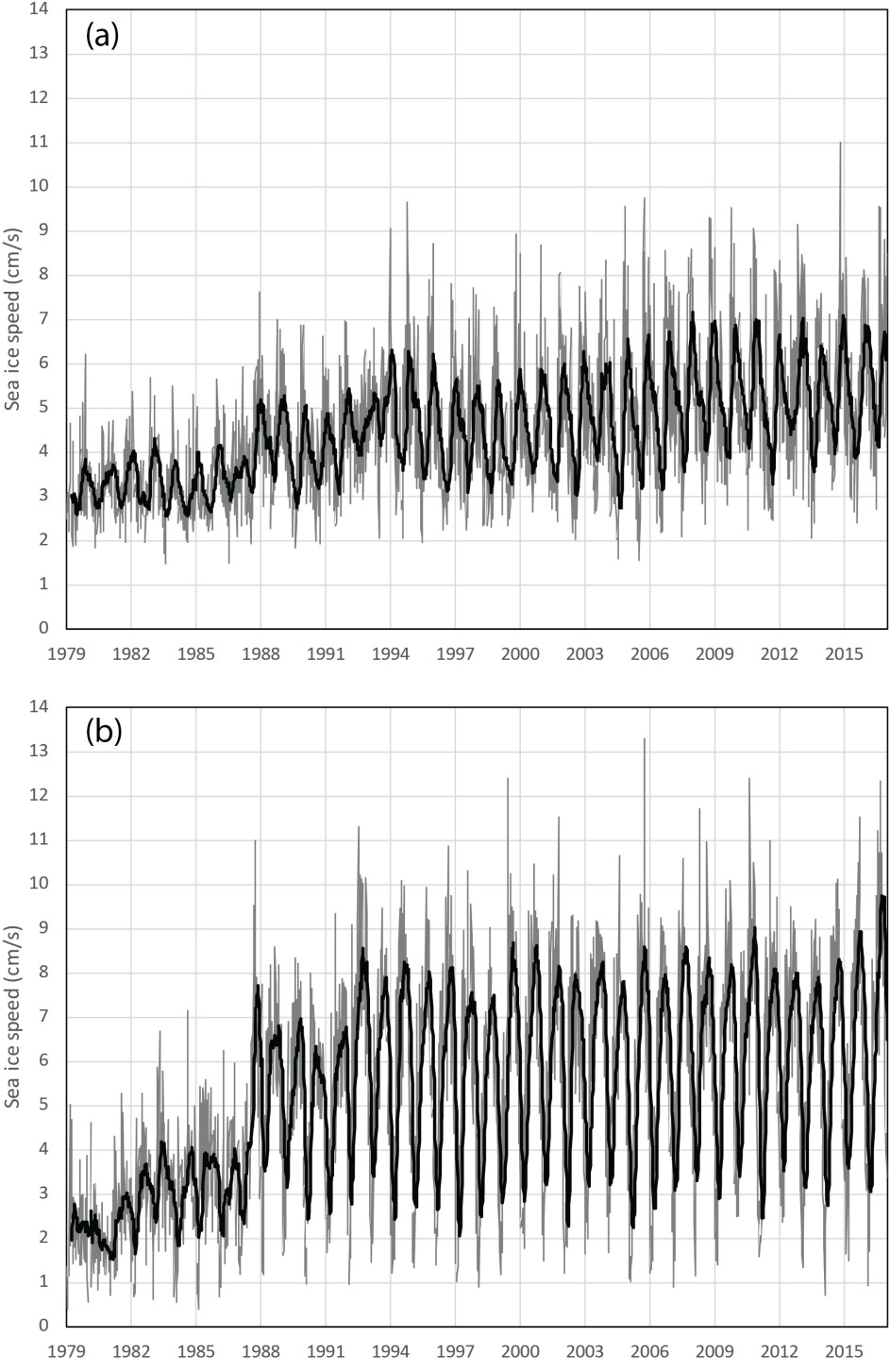

**Figure 8.** Arctic weekly average sea ice drift speed for Version 4, 1979-2017. A 13-week running average is overlaid on the weekly values to highlight seasonal variability.

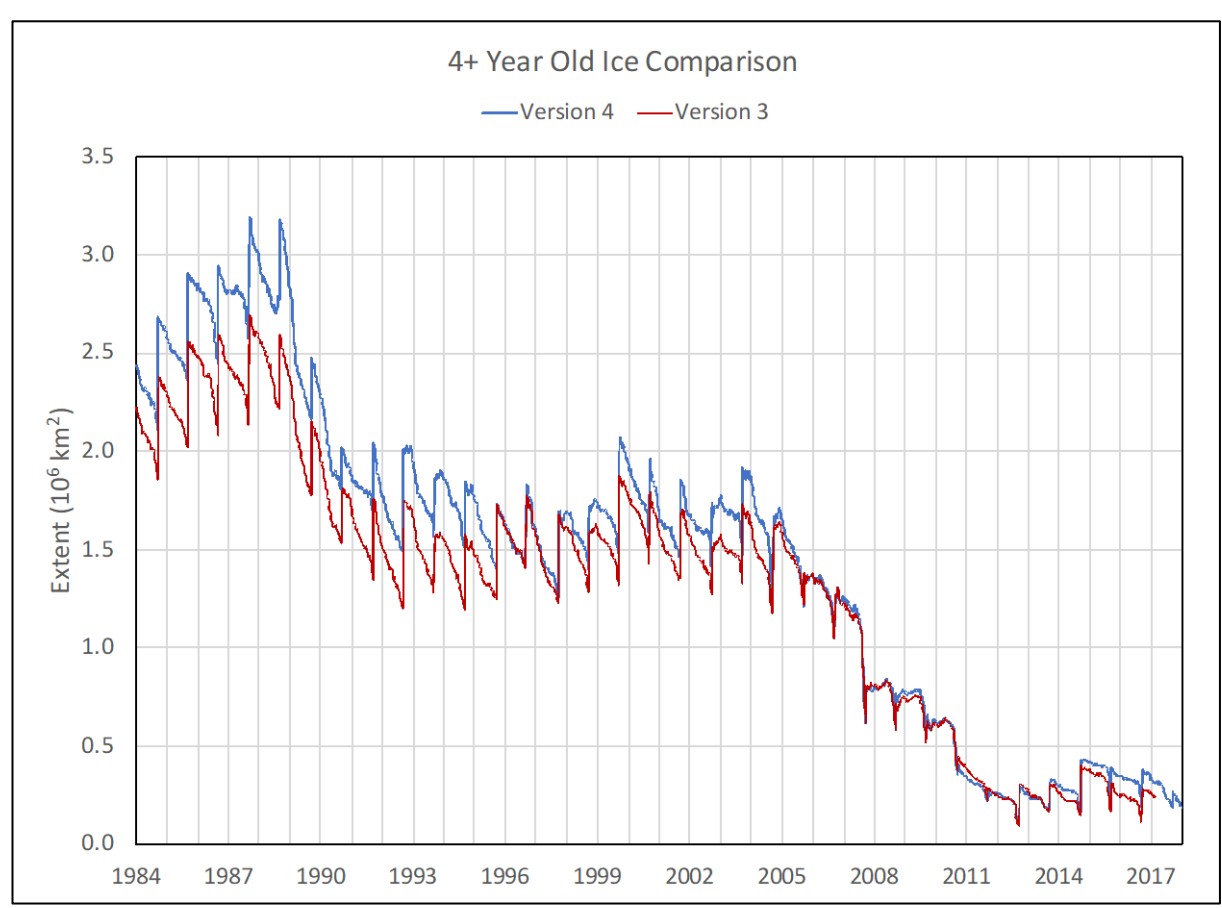

**Figure 9.** Comparison of 4+ year old ice from Version 3 (red) and Version 4 (blue) for 1984-2017.

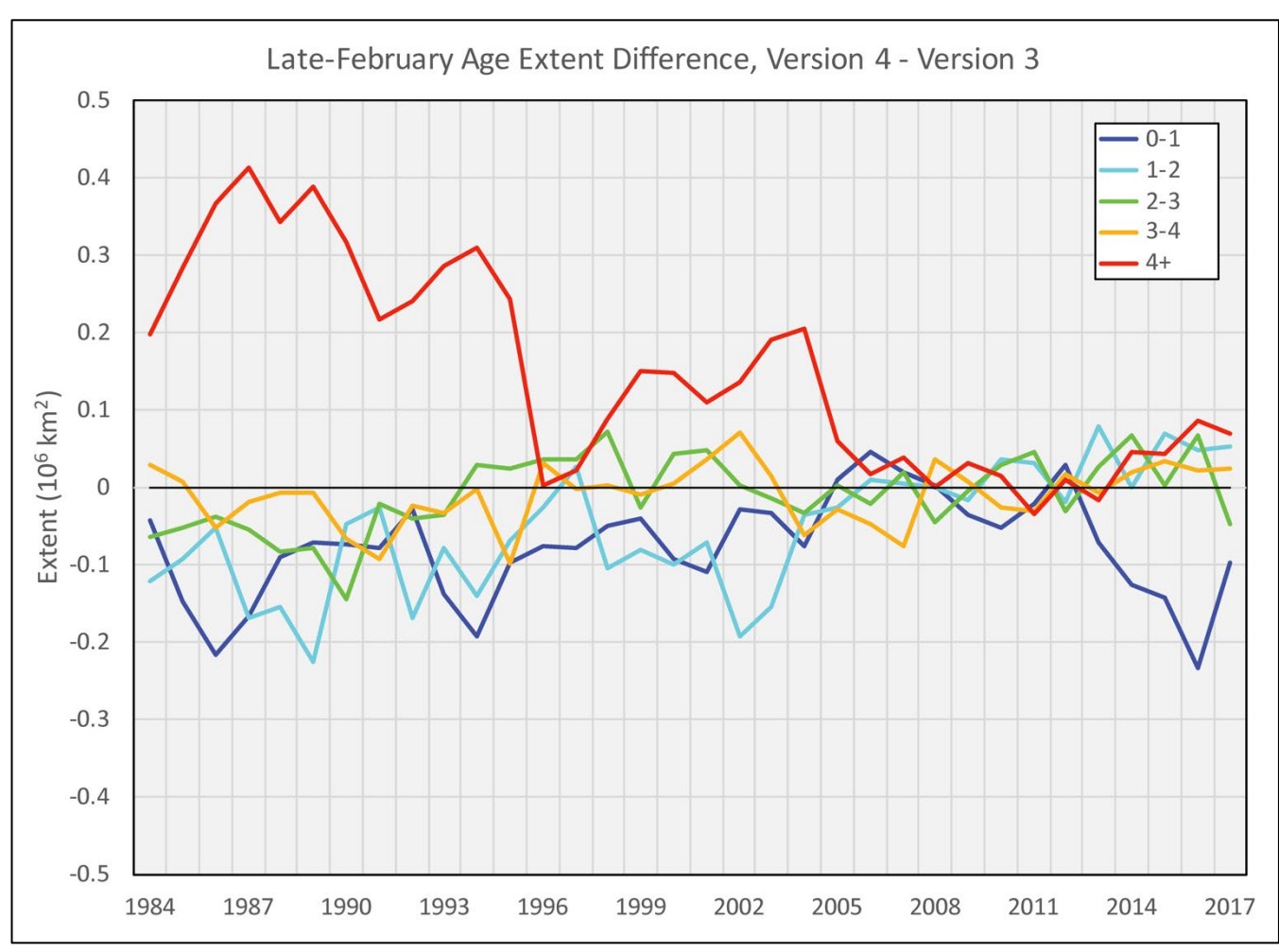

**Figure 10.** Extent difference between Version 4 and Version 3 sea ice age categories for the week of February 19-25 from 1984 to 2017.

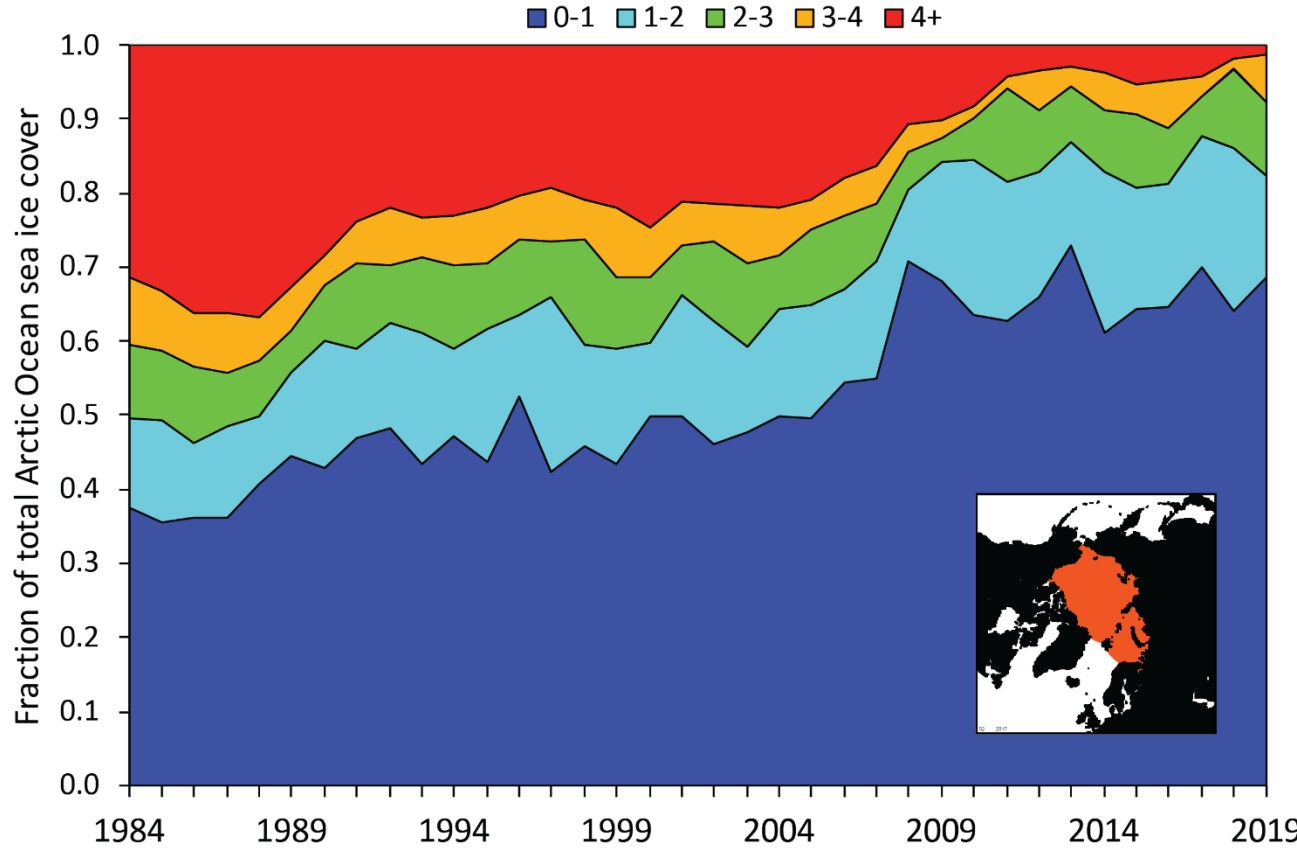

**Figure 11.** Timeseries of fraction of total sea ice coverage by sea ice age category for the week of February 19-25, 1984-2019. These timeseries are for the Arctic Ocean region, which is the region shaded in orange in the lower right inset image (Perovich et al., 2019), used to include only regions were MYI may exist at a non-negligible level.

