# Peer review of "An enhancement to sea ice motion and age products at NSIDC"

_The Cryosphere, 2019_

## Referee Comment (RC1) · Anonymous Referee #1 · 24 Apr 2019

The paper provides a detailed and very well written description of the new ice motion and ice age product to be delivered by NSIDC. Importance of these products is justified by a comprehensive introduction. Changes in the production chain at all stages - from individual drift components to ice age computation - are properly documented. It is illustrated that the changes at the lower level (new optimal interpolation scheme) have impact at the higher level products (larger extent of older ice) but predominantly in the beginning of the observation period (before 1996).

Notwithstanding the high quality of the paper, in my opinion it fails to quantitatively prove that the ice motion and age products have been actually enhanced. The only evidence that ice motion was improved is qualitative - visual comparison of drift components on figures 3 and 4. Improvement of the ice age product is also illustrated only

visually - a more homogeneous ice age distribution is presented on figure 6. Given the high demand for these products a proper quantitative validation is of vital importance. A section needs to be added where the ice motion is compared with other existing independent ice motion products including, for example, AMSR2 derived drift, SAR derived drift, drift of the buoys that were excluded from the optimal interpolation, etc. Although a direct validation of ice age product is probably impossible due to absence of a similar independent product, it can be validated indirectly by comparison of multi-year ice extent with products derived from passive microwave sensors or scatterometers in March - April. It is required to include in this section the widely used product quality metrics such as RMSE, bias, Pearson correlation coefficient, etc. (and preferably both for version 3 and version 4) in order quantitatively prove the enhancement of the products and illustrate applicability in different scientific domains (trend computation, assimilation in numerical models, etc).

Minor comments No grammar mistakes or typos were identified and the minor comments only concern few clarifications / corrections that are needed in the text.

P3, L23 and L25. Some authors distinguish feature-tracking (detection of individual keypoints on two images => description of keypoints by a binary vector based on => brute-force matching of keypoints, eg. SURG, ORB, etc) from pattern-matching (maximum cross correlation continuously applied to every n-th pixel) [e.g. Rublee et al., 2011, Berg et al., 2014, Korosov et al., 2017]. Maybe a consistent use of "pattern-matching" is preferable in these two cases.

P4, L18. How was the effectiveness of 4X oversampling estimated?

P5, L5. What is the criterion for omitting rogue vectors? Difference from median of vectors in the vicinity? What is the threshold for screening?

P5, L12. What were the thresholds used in V3 and V4 for filtering PMW vectors?

P5, L22. 1% seem to be quite an underestimation of ice drift speed. In addition,

this relation cannot be constant in space and time. With the available large amount of collocated data on wind speed and observed ice drift it should be quite simple to illustrate validity of this 'constant 1%' assumption. It would be important to justify it, e.g. in the Discussion section where the relationship between ice drift and wind speed is illustrated spatially and temporally.

P7, L12. Is there a proof that the motion is "largely unbiased"? It is important to add a validation section (as explained in the general comment section above) to prove this statement.

P8, L5 and L6. What is the impact of values of C and D parameters on the drift speed quality (visual appearance) and accuracy (as can be retrieved from validation)? How sensitive are the motion and age products?

P9, L26. I'm confused by the phrase "...all parcels in the 12.5 km ice age grid are initialized with an age-class...". Does it mean that there are several parcels per grid cell? How many?

P10, L19. How much "substantially"? It would be nice to have a numerical characteristics to compare V3 and V4.

P11, L7. I don't quite agree that the difference between V3 and V4 is "fairly consistent over time". It grows from almost 0 (between 1980 and 1986) to almost 1 cm/s (between 2012 and 2017)! It clearly contributes to the difference in drift speed trends between v3 and v4. But which one is more correct? It is very unfortunate that proper quantitative validation is not provided. Maybe this difference is an indication of uncertainty of the motion product and the observed trends are actually statistically insignificant?

---

## Referee Comment (RC2) · Anonymous Referee #2 · 1 Jul 2019

Review of "An enhancement to sea ice motion and age products" by Mark A. Tschudi et al. TC-2019-40

General comments: 1) 7-10: Stale opening sentences in the abstract. Reads boring, repeats phrases.

2) Abstract: Suggest to provide more "scientific" results/summary. And drop the first few sentences.

3) The ms is a bit plain and could be lifted by addition of further investigation of the ice-motion and ice-age data sets and discussion of the results.

Specific comments:

[Figure]

1/12-13: Pls specify/give example on how they "are not substantially different between the versions."

1/18: "recent years" or "recent decades"?

1/26: Suggest to rephrase "it is more difficult to draw solid quantitative".

2/26: Correct "will expand greatly with the launch of the NASA ICESat-2 in September 2018" as all this is happened (i.e., it is not longer in the past).

2/27: Could mention Op IceBridge in this paragraph.

3/8-9: Redundant?

3/12-14: Shorten.

3/17: Change "ice motion" to "sea ice motion".

3/25-26: Provide info on typical repeat frequency of "Two geolocated, spatially-coincident, temporally-consecutive satellite images".

4/18: How is the oversampling rate of "4" motivated?

4/34: This statement is not correct as is: "AVHRR was discontinued after 2000." Please qualify or remove.

5/3: How is the threshold of "0.4" motivated?

5/8-15: It is not clear how exactly previous versions dealt with input PM data. Can you separate into composite versus swath or similar?

5/19-24: The assumption that sea ice moves at 0.01 of the wind speed (for the Arctic) needs to be reviewed, especially in an environment of highly variable and increasing wind speeds. –> Underestimate of the ice speed. I.e., Rampal et al. [2009], Positive trend in the mean speed and deformation rate of Arctic sea ice, 1979–2007, J. Geophys. Res., 114, C05013, doi:10.1029/2008JC005066.

5/29: Replace "data" in "These buoys monitor meteorological and oceanographic data", i.e., to read "conditions" or "states".

5/33: Mention explicitly that there are too few sea-ice buoys in the Southern Ocean.

8/23ff: It is not clear how the few PM (or combined) motion vectors are treated to derive a broad map of sea-ice motion (on EASE grid)? It appears as if severe extrapolation is taking place.

8/24: There are several experiments with decent buoy arrays available for some parts of the Antarctic sea-ice zone. Why not use some of those to at least assess the skill of the product... and to possibly explore the suitability of Antarctic ice-buoy data to provide information into the ice-motion product discussed here.

9/2: The netCDF file should include an additional mask (0/1) where one can mask all gridded ice motion that is "too far" from an actual observation, where the value of "too far" needs to be discussed.

9/32: How is the limit of "16 years" for the maximum ice age set? Physical motivation?o

10/19: There is not quantitative measure of how V4 ice age as improved relative to V3: "there is less "speckling"".

10/21ff: In discussing the relative "ageing" of Arctic ice from V3 to V4 there are no physical details provided as to what process would be the main driver of this change.

11/1ff: The discussion of trends and variability in ice motion & age between V3 and V4 should be more quantitative. – Also, regional contributions should be explored.

12/7: Correct "Fennoscandian peninsula." to "Fennoscandian Peninsula." (upper case)

Fig.7: There seems to be a cyclical signal in the ice-speed difference between V3 and V4. Decadal or perhaps 11 - 12 years. Can different PM sensors be the reason for this? Or the speed magnitude??

Fig.9: The version difference in ice age for 4yr+ is not well explained.

Fig.10 & 11 are not well explained/discussed.

---

## Referee Comment (RC3) · Anonymous Referee #3 · 3 Jul 2019

Review of

An enhancement to sea ice motion and age products

by

Tschudi, M., et al.

Summary: This contribution attempts to illustrate the enhancements - to be understood mainly as extension - of two sea-ice products issued by the National Snow and Ice Data Center (NSIDC), namely the NSIDC sea-ice motion data set and the NSIDC sea-ice age data set. The latter is based on the former. The manuscript advertizes the data sets, informs a bit about the history of these two data sets and describes briefly changes made to the processing which potentially led to an enhancement in quality of

both products.

My overall impression is that this paper is not suitable for publication and should be rejected.

It lacks essential information about the retrieval procedure, and the retrieval uncertainties. It further lacks results of an evaluation. It presents trends which seem artificial. It is incomplete in terms of geographical coverage. It contains errors. There are many open technical questions which are not answered in the manuscript and also not in the respective documentation of the data set(s) on the NSIDC web pages.

This paper is written as if it extends a reference benchmark paper where all the required missing details could be found. But this is not the case. Such a benchmark paper does not yet exist for the sea-ice motion product. As the authors stress, the sea-ice motion data set is unique, it has a unique length, and it allows unique applications. And therefore it requires a unique extensive high-quality paper first, in which the reader and the data users can learn about all details and limitations associated with the data set and its generation and evaluation.

General Concerns: GC1: No systematic evaluation of the products has been undertaken - neither for version 3 nor for version 4 of the sea-ice motion product. Also the associated newest sea-ice age data set is not evaluated. In your case, it is not sufficient to just compare version 3 and version 4 of the product because a systematic, detailed evaluation of version 3 products is missing in the scientific literature. There is hence no benchmark against which this new version 4 can be quantitatively referenced. Section 2.2 does not provide new results. There is no indication of a useful sea-ice motion retrieval uncertainty provided along with the product, like is done for sea-ice concentration and thickness data sets. The authors do not present results of an evaluation neither of the newly derived components of the sea-ice motion entering the gridded product nor of the gridded product.

GC2: The reader and data set user is informed about user statistics, the importance

of the two data sets, some selected bits of the history of the retrievals, and a relatively unspecific description of the changes made to the methods which leaves many open questions. This is, however, potentially not what a reader of this paper and user of this data set would have expected for the following reason: There is no specific paper in which the various retrieval processes, their uncertainties, the caveats of the different spatio-temporal resolution of the input data sets, a detailed description of the merging (optimal interpolation) approach and its uncertainties have been published so that the full package of detailed, high-quality information is visible at a glance. The retrieval, the input data, the pre-processing steps all these are not transparently described. In other words: A benchmark reference paper containing all bits and pieces is missing so far. And in this context this paper about an "enhancement" seems of doubtful value.

GC3: The introduction is a nice compilation of recent work dedicated to changes in Arctic sea-ice area / extent, multiyear ice fraction and thickness and in Antarctic sea-ice area and extent. But: during the past two decades or so various other approaches for sea-ice motion retrieval have been developed and the respective data sets are also in use. This paper lacks a review of this work. The retrieval method is not put into context of the current research landscape in this field. This applies to new algorithm developments (both method and input satellite data) as well as evaluation studies. What I, in this context, understand the least, is that despite evidence exists in the literature from various groups using predecessors of the ice motion data set (mainly version 2 and 3), that the inter-sensor inconsistencies cause artificial trends computed from the ice motion product and render parts of the product not useful, you do not comment about this.

GC4: I also miss an evaluation of the ice-age data set and/or more quantitative statements about its reliability and potential uncertainty. I do not rate a comparison to the previous version of the data set as providing enough evidence for a proven enhancement. Such a comparison provides only qualitative information about the potential sign of the enhancement. Any quantitative information which would go beyond the comparison to the previous version is lacking.

There is a list of specific comments which I can provide on request if need be.

---

## Referee Comment (RC4) · Bruno Tremblay (Referee) · 9 Jul 2019

Title: An enhancement to sea-ice motion and age products
Author: M. Tschudi, W. Meier, S Stewart
* * *
This paper present a complete documentation of the Polar Pathfinder sea ice drift and age dataset hosted on the National Snow and Ice Data Center as well as the improvement made from Version 3 to Version 4. The main improvement in the drift dataset is in the optimal interpolation scheme used to merge the satellite, buoy and free-drift estimates into one single dataset. The new scheme now use a weighted average of different drift products, with the weight calculated from their respective errors, and a radius of influence that is based on the decorrelation spatial scale derived from observations. The main improvement in the age dataset is the use of the new ice drift dataset discussed in this paper. Main results include a significant reduction in large spatial gradients at the junction where buoy and satellite products were used in the merged product (mainly in the earlier part of the record where buoy data is more scarce), faster sea ice drift speed and more older ice again in the earlier part of the record (when compared with Version 3).

This paper was long awaited. As stated by the authors, the NSIDC drift and age data is used by several groups, but was lacking a single source in the scientific literature that can be cited describing in details the method used to create the dataset. The paper is mostly well written except in places where some editing is required (see example below).

I recommend that the paper be published after addressing the comments below.

Comments:

Title: Mention polar pathfinder and/or NSIDC in the title.

Page 3, line 13: "…along as well as nothing the changes…". This sentence needs to be rephrased.

Page 3, line 16: Include a table listing all data sources, the time period when the data is used in the merged product and the spatial resolution of the data product.

Page 3, line 19: It should be mentioned here that the optimal interpolation scheme is described in more details below, so the reader knows there is more than just Figure 1 describing the new scheme.

Page 4, line 19: 7.23 cm/sec is still larger than the mean sea ice drift in the Beaufort Gyre. It also means that the satellite drift estimates are not continuous. Later the authors mention that temporal interpolation into a weekly product and spatial (optimal) interpolation smooths the derived velocity field. A sentence should added to tell the reader that this is discussed later on in the paper.

Page 4, line 19: The over-sampling procedure need to be described. Are the images linearly interpolated to get this sub-pixel resolution? If so, what kind of interpolation is used? I would think that the error in the over-sampled images would be a function of the interpolation scheme (linear versus non-linera), etc. Please discuss. Also, how large is the window that is translated to get at the maximum correlation?

Page 4, line 33: Please rephrase in the active form for clarity.

Page 5, line 11: "… and there was NO provenance…" instead?

Page 5, line 19: The spatial resolution of the wind product needs to be stated.

Page 6, line 3: This corresponds to a weighted average drift speed for a 36-hour period with weight 1, 2, 1 for the midnight-noon, noon-midnight, and midnight-noon (next day) drift estimates. Or are these calculated as (2*Drift(midnight-midnight) + 1*Drift(Noon-Noon) ) / 3 for a true 1-day average? This should be clarified

Page 6, line 33: Write "cancel each other" instead.

Page 6, line 4: Give the bias estimates for all products quoted in this paragraph, in a similar manner as for the SSMI daily velocity component (Meier et al, 2000.

Page 7, line 14: Is the SHEBA GPS data part of the IABP buoy dataset? If so, we do not expect a large error because it is used in the polar path finder dataset.

Page 8, line 1: With fixed C values, the sum of all weights used for a given estimate will not add up to 1. An additional formula used to calculate the final (optimally interpolated) drift velocity (including the division by the sum of the weights) should be added.

Page 8, line 1: The error for each drift product (and used to calculate C) should be included in a Table.

Page 8, line 2: I am guessing that D will depend on the satellite product used to calculate the spatial cross-correlation, when in reality, the de-correlation length scale would be only location dependent. This should be clarified. Please state whether D is a constant or if it varies spatially.

Page 8, line 13: Interpolating or averaging? If interpolating, what interpolation scheme (i.e. bi-linear interpolation, kriging, spline) is used?

Page 9, line 5: This is the first time mentioned. What is the resolution of the daily product if not the same as the weekly product?

Page 9, line 6: It is desirable to use the weekly product for all applications, no? I.e. not just for trend estimates.

Page 9, line 21: Put "in year" in parenthesis at the end of sentence for better flow.

Page 10, line 9: Error in drift vectors can lead to convergence of different track into the same grid cell. I believe that the oldest of the tracks is retained in the algorithm. This is another mechanism by which younger ice is lost. It should be mentioned here.

Page 10, line 10: "to spin-up to obtain". Please rephrase.

---

## Author Comment (AC1) · 3 Aug 2019

To the editor and reviewers of The Cryosphere regarding manuscript tc-2019-40:

Please note that this same document has been supplied to each reviewer. Our comments often include more than one reviewer, so it was best for all to see the same response.

**General comments and request for extension to implement an update plan:**

Thank you for obtaining four reviews for our paper. These will improve the quality of our paper.

Per the editor's request, we have not prepared a revised manuscript at this time. In order to respond to all of the issue raised by referees, we will significantly revise the manuscript. We generally classify the referees' comments as major and minor. The minor comments will be addressed by appropriate additions and clarifications to the paper. The major comments require additional analysis on our part. We will do this new work and incorporate the findings into the manuscript.

We will need a bit of time to submit this revised paper. Given that there was a long delay in receiving comments, our planned revision schedule shifted to later in the year. We propose to complete our update to the manuscript by the end of September.

This document outlines our plan to respond to all reviewers' comments, both major and minor. If this plan is acceptable to you, we will implement it by the end of September.

**Authors' response to the range of referee recommendations:**

The reviewers have a wide range of recommendations: one requested minor revisions, two requested major revisions, and one recommended rejection.

In addition to validation issues noted by Referees #1 and #2, the rejection recommended by Referee #3 seems to be based primarily on a lack of peer-reviewed documentation of earlier versions. We recognize this issue and agree that it would be better if fuller peer-reviewed documentation had been published earlier, but that was the province of the original PIs of the product.

Now that we—the authors of the present manuscript—have taken over the production, we have worked to improve documentation of the product and by submitting this manuscript, we are seeking a peer-reviewed publication that provides both a comprehensive description of the data set and a specific description of the changes in the product as it was updated from version 3 to version 4. Several specific concerns in the literature – e.g., the problem noted by Szanyi et al. regarding the non-optimal weighting of the buoys that led to "bulls-eye" patterns in the daily motions – are addressed with the version 4 update. With the additional validation and description of the data set that we outline below, it is our intention—also noted by Referees #1 and #4—that this paper shall become the definitive peer-reviewed reference for this data set.

Referee #3 is also critical of the level of documentation for this product. We note that both Referee #1 and #4 disagree with this assessment and are complimentary of the documentation that we included in the manuscript. We note that further details on the basic processing are included in the online NSIDC documentation for the products. We feel the documentation is on par with many other published data sets and is generally sufficient. We also note that several peer-reviewed papers have been published

using the products to investigate science products and that the ice age product has regularly been included in the NOAA Arctic Report Card, which includes a peer review.

If there are some specific areas that you feel are lacking in documentation, we can certainly try to address them. If there are any other specific issues raised by Referee #3 that you feel we need to address, please let us know.

**Overview of referee comments:**

We have included all comments by all reviewers at the end of this document. Those reviews are color-coded by reviewer. Major comments are highlighted. We judge all other comments to be minor comments, all of which we will address.

Considering all reviewers' comments together, we observe three primary major concerns in common to most of the reviews:

- **Major Concern 1**: The first is the lack of quantitative validation of the motion and age estimate.
- **Major Concern 2**: The second is in how the difference between version 3 and version 4 changes over time.
- **Major Concern 3**: The third is the use of a 1% rule for deriving ice motion from wind speed.

Here, we outline how we plan to address each of these major comments.

**Major Concern 1** specifically refers to issues of quantitative validation.

The impetus for this manuscript is primarily to demonstrate that we've improved the processing – particularly in regards to the interpolation of the buoys -- and thus our focus was simply to show the differences between the versions. However, we recognize that the reviewers have legitimate points on showing some quantitative validation. For instance, it will be helpful to compare both sea ice motion and sea ice age to independent observations or proxies.

We plan to add a section on quantitative validation to the paper. For ice motion, we will use independent buoys to derive motion uncertainty statistics (bias, RMSE, etc.) to quantify the difference in uncertainties in V3 and V4. This will be focused on the Arctic. The Antarctic ice motion estimates are considered to be more experimental, and motions there are known to be of lower quality. If we find suitable buoy observations to provide some error quantification with a reasonable amount of effort, we will consider doing that. However, this is lower priority than other issues in our view.

As Referee #1 notes, quantitative validation of sea ice age is not possible. However, we may be able to at least compare first-year vs. multi-year sea ice coverage. This at least quantifies the reduction in FYI bias due to unrealistic convergence noted in Szanyi et al. We will do this via comparison with U.S. National Ice Center sea ice charts.

**Major Concern 2** refers to questions of how the data set in version 3 and 4 change over time

The validations outlined above will also help us address the second main comment– how the differences between V3 and V4 change over time. We believe this is primarily due to the change to the buoy interpolation since that was the most significant change. The uncertainty quantification approach noted above will indicate improvement from V3 to V4. So, differences indicate an improvement in V4. If independent buoys exist during requisite periods, we can also compare different years at different periods in the record to quantify how the changes from V3 to V4.

**Major Concern 3** refers to parameters used in one of the algorithms used to compute a component of the data set.

The final point is in regards to the use of the 1% rule for wind-derived ice motions. We recognize that this is likely not optimal, particularly with increasing speed. The 1% factor is based on peer-reviewed literature (albeit rather old) and was defined in the product by the earlier PIs. We kept that the same for simplicity while focusing on higher priority issues. The wind-forced motions have low weighting in the OI scheme and generally have a limited influence on the overall motion product. We will do a simple sensitivity test to show the effect of using 1% vs. a more physically realistic 1.5-2%.

**Response summary:**

We will incorporate our responses to each major comment by either adding new sections to the paper, updating existing sections, or providing an appropriately detailed supplement to the manuscript.

As noted above, we will also address all reviewers' minor comments by clarifying or improving the noted portions of the original manuscript.

Please let us know if you find our plan satisfactory and if we have missed any major concerns that you feel we need to address. Also, please let us know if our proposed time line for submitting a revised manuscript by the end of September is acceptable.

Regards –
Mark Tschudi, Walt Meier, and J. Scott Stewart

For reference, the comments of the four referees for The Cryosphere manuscript "tc-2019-40" are provided here.

Comments are color-coded per referee, and those we identify as major are ==highlighted==.  Our authors' responses below are preceded by "AR:".

Reviewer #1:

> The paper provides a detailed and very well written description of the new ice motion and ice age product to be delivered by NSIDC. Importance of these products is justified by a comprehensive introduction. Changes in the production chain at all stages - from individual drift components to ice age computation - are properly documented. It is illustrated that the changes at the lower level (new optimal interpolation scheme) have impact at the higher level products (larger extent of older ice) but predominantly in the beginning of the observation period (before 1996).

> Notwithstanding the high quality of the paper, in my opinion ==it fails to quantitatively prove that the ice motion and age products have been actually enhanced. The only evidence that ice motion was improved is qualitative - visual comparison of drift components on figures 3 and 4. Improvement of the ice age product is also illustrated only visually - a more homogeneous ice age distribution is presented on figure 6. Given the high demand for these products a proper quantitative validation is of vital importance. A section needs to be added where the ice motion is compared with other existing independent ice motion products== including, for example, AMSR2 derived drift, SAR derived drift, drift of the buoys that were excluded from the optimal interpolation, etc. Although a direct validation of ice age product is probably impossible due to absence of a similar independent product, it can be validated indirectly by comparison of multi-year ice extent with products derived from passive microwave sensors or scatterometers in March - April. ==It is required to include in this section the widely used product quality metrics such as RMSE, bias, Pearson correlation coefficient, etc. (and preferably both for version 3 and version 4) in order quantitatively prove the enhancement of the products and illustrate applicability in different scientific domains (trend computation, assimilation in numerical models, etc).==

AR: Our additional validation work and consequent comparisons between versions 3 and 4 will address these concerns.

> Minor comments No grammar mistakes or typos were identified and the minor comments only concern few clarifications / corrections that are needed in the text.

> P3, L23 and L25. Some authors distinguish feature-tracking (detection of individual keypoints on two images => description of keypoints by a binary vector based on => brute-force matching of keypoints, eg. SURG, ORB, etc) from pattern-matching (maximum cross correlation continuously applied to every n-th pixel) [e.g. Rublee et al., 2011, Berg et al., 2014, Korosov et al., 2017]. Maybe a consistent use of "pattern-matching" is preferable in these two cases.

AR: We will reference different feature-tracking methods.

P4, L18. How was the effectiveness of 4X oversampling estimated?

P5, L5. What is the criterion for omitting rogue vectors? Difference from median of vectors in the vicinity? What is the threshold for screening?

P5, L12. What were the thresholds used in V3 and V4 for filtering PMW vectors?

AR: We will update the text to address these issues.

P5, L22. 1% seem to be quite an underestimation of ice drift speed. In addition, this relation cannot be constant in space and time. With the available large amount of collocated data on wind speed and observed ice drift it should be quite simple to illustrate validity of this 'constant 1%' assumption. It would be important to justify it, e.g. in the Discussion section where the relationship between ice drift and wind speed is illustrated spatially and temporally.

AR: We will describe the effect of using a larger wind-ice speed relationship.

P7, L12. Is there a proof that the motion is "largely unbiased"? It is important to add a validation section (as explained in the general comment section above) to prove this statement.

P8, L5 and L6. What is the impact of values of C and D parameters on the drift speed quality (visual appearance) and accuracy (as can be retrieved from validation)? How sensitive are the motion and age products?

P9, L26. I'm confused by the phrase "...all parcels in the 12.5 km ice age grid are initialized with an age-class...". Does it mean that there are several parcels per grid cell? How many?

P10, L19. How much "substantially"? It would be nice to have a numerical characteristics to compare V3 and V4.

P11, L7. I don't quite agree that the difference between V3 and V4 is "fairly consistent over time". It grows from almost 0 (between 1980 and 1986) to almost 1 cm/s (between 2012 and 2017)! It clearly contributes to the difference in drift speed trends between v3 and v4. But which one is more correct? It is very unfortunate that proper quantitative validation is not provided. Maybe this difference is an indication of uncertainty of the motion product and the observed trends are actually statistically insignificant?

AR: We will update the text to address these issues.

Reviewer #2:

Review of "An enhancement to sea ice motion and age products" by Mark A. Tschudi et al. TC-2019-40

General comments:

1) 7-10: Stale opening sentences in the abstract. Reads boring, repeats phrases.

2) Abstract: Suggest to provide more "scientific" results/summary. And drop the first few sentences.

3) The ms is a bit plain and could be lifted by addition of further investigation of the ice-motion and ice-age data sets and discussion of the results.

AR: We will update the text to address these issues.

Specific comments:

1/12-13: Pls specify/give example on how they "are not substantially different between the versions."

1/18: "recent years" or "recent decades"?
1/26: Suggest to rephrase "it is more difficult to draw solid quantitative".

2/26: Correct "will expand greatly with the launch of the NASA ICESat-2 in September 2018" as all this is happened (i.e., it is not longer in the past).

2/27: Could mention Op IceBridge in this paragraph.

3/8-9: Redundant?

3/12-14: Shorten.

3/17: Change "ice motion" to "sea ice motion".

3/25-26: Provide info on typical repeat frequency of "Two geolocated, spatially-coincident, temporally-consecutive satellite images".

4/18: How is the oversampling rate of "4" motivated?

4/34: This statement is not correct as is: "AVHRR was discontinued after 2000." Please qualify or remove.

5/3: How is the threshold of "0.4" motivated?

5/8-15: It is not clear how exactly previous versions dealt with input PM data. Can you separate into composite versus swath or similar?

AR: We will update the text to address these issues.

5/19-24: The assumption that sea ice moves at 0.01 of the wind speed (for the Arctic) needs to be reviewed, especially in an environment of highly variable and increasing wind speeds. –> Underestimate of the ice speed. I.e., Rampal et al. [2009], Positive trend in the mean speed and

deformation rate of Arctic sea ice, 1979–2007, J. Geophys. Res., 114, C05013, doi:10.1029/2008JC005066.

AR: We will present an analysis of overall motion response to an increase in the wind-ice speed relationship.

5/29: Replace "data" in "These buoys monitor meteorological and oceanographic data", i.e., to read "conditions" or "states".

5/33: Mention explicitly that there are too few sea-ice buoys in the Southern Ocean.

8/23ff: It is not clear how the few PM (or combined) motion vectors are treated to derive a broad map of sea-ice motion (on EASE grid)? It appears as if severe extrapolation is taking place.

AR: We will update the text to address these issues.

8/24: There are several experiments with decent buoy arrays available for some parts of the Antarctic sea-ice zone. Why not use some of those to at least assess the skill of the product... and to possibly explore the suitability of Antarctic ice-buoy data to provide information into the ice-motion product discussed here.

AR: We will use available Antarctic buoys to provide additional validation of the Southern Hemisphere motions.

9/2: The netCDF file should include an additional mask (0/1) where one can mask all gridded ice motion that is "too far" from an actual observation, where the value of "too far" needs to be discussed.

9/32: How is the limit of "16 years" for the maximum ice age set? Physical motivation?o

10/19: There is not quantitative measure of how V4 ice age as improved relative to V3: "there is less "speckling"".

10/21ff: In discussing the relative "ageing" of Arctic ice from V3 to V4 there are no physical details provided as to what process would be the main driver of this change.

11/1ff: The discussion of trends and variability in ice motion & age between V3 and V4 should be more quantitative. – Also, regional contributions should be explored.

12/7: Correct "Fennoscandian peninsula." to "Fennoscandian Peninsula." (upper case)

Fig.7: There seems to be a cyclical signal in the ice-speed difference between V3 and V4. Decadal or perhaps 11 - 12 years. Can different PM sensors be the reason for this? Or the speed magnitude??

Fig.9: The version difference in ice age for 4yr+ is not well explained.

Fig.10 & 11 are not well explained/discussed.

AR: We will update the text to address these issues.
* * *
Review of An enhancement to sea ice motion and age products by Tschudi, M., et al.

Summary: This contribution attempts to illustrate the enhancements - to be understood mainly as extension - of two sea-ice products issued by the National Snow and Ice Data Center (NSIDC), namely the NSIDC sea-ice motion data set and the NSIDC sea-ice age data set. The latter is based on the former. The manuscript advertizes the data sets, informs a bit about the history of these two data sets and describes briefly changes made to the processing which potentially led to an enhancement in quality of both products.

My overall impression is that this paper is not suitable for publication and should be rejected.

It lacks essential information about the retrieval procedure, and the retrieval uncertainties. It further lacks results of an evaluation. It presents trends which seem artificial. It is incomplete in terms of geographical coverage. It contains errors. There are many open technical questions which are not answered in the manuscript and also not in the respective documentation of the data set(s) on the NSIDC web pages.

This paper is written as if it extends a reference benchmark paper where all the required missing details could be found. But this is not the case. Such a benchmark paper does not yet exist for the sea-ice motion product. As the authors stress, the sea-ice motion data set is unique, it has a unique length, and it allows unique applications. And therefore it requires a unique extensive high-quality paper first, in which the reader and the data users can learn about all details and limitations associated with the data set and its generation and evaluation.

AR: We expect that our additional validation work will improve the manuscript so that it serves as the benchmark paper for this data set.

General Concerns:

GC1: No systematic evaluation of the products has been undertaken - neither for version 3 nor for version 4 of the sea-ice motion product. Also the associated newest sea-ice age data set is not evaluated. In your case, it is not sufficient to just compare version 3 and version 4 of the product because a systematic, detailed evaluation of version 3 products is missing in the scientific literature. There is hence no benchmark against which this new version 4 can be quantitatively referenced. Section 2.2 does not provide new results. There is no indication of a useful sea-ice motion retrieval uncertainty provided along with the product, like is done for sea-ice concentration and thickness data sets. The authors do not present results of an evaluation neither of the newly derived components of the sea-ice motion entering the gridded product nor of the gridded product.

AR: Our additional validation work will permit this evaluation.

GC2: The reader and data set user is informed about user statistics, the importance of the two data sets, some selected bits of the history of the retrievals, and a relatively unspecific description of the changes made to the methods which leaves many open questions. This is, however, potentially not what a reader of this paper and user of this data set would have expected for the following reason: There is no specific paper in which the various retrieval processes, their uncertainties, the caveats of the differentspatio-temporal resolution of the input data sets, a detailed description of the merging (optimal interpolation) approach and its uncertainties have been published so that the full package of detailed, high-quality information is visible at a glance. The retrieval, the input data, the pre-processing steps all these are not transparently described. In other words: A benchmark reference paper containing all bits and pieces is missing so far. And in this context this paper about an "enhancement" seems of doubtful value.

AR: Our additional validation work will improve the manuscript so that it serves as the benchmark paper for this data set.

GC3: The introduction is a nice compilation of recent work dedicated to changes in Arctic sea-ice area / extent, multiyear ice fraction and thickness and in Antarctic sea-ice area and extent. But: during the past two decades or so various other approaches for sea-ice motion retrieval have been developed and the respective data sets are also in use. This paper lacks a review of this work. The retrieval method is not put into context of the current research landscape in this field. This applies to new algorithm developments (both method and input satellite data) as well as evaluation studies. What I, in this context, understand the least, is that despite evidence exists in the literature from various groups using predecessors of the ice motion data set (mainly version 2 and 3), that the inter-sensor inconsistencies cause artificial trends computed from the ice motion product and render parts of the product not useful, you do not comment about this.

AR: The artificial divergence issues raised by Szyani et al. will be specifically addressed.

GC4: I also miss an evaluation of the ice-age data set and/or more quantitative statements about its reliability and potential uncertainty. I do not rate a comparison to the previous version of the data set as providing enough evidence for a proven enhancement. Such a comparison provides only qualitative information about the potential sign of the enhancement. Any quantitative information which would go beyond the comparison to the previous version is lacking.

There is a list of specific comments which I can provide on request if need be.

AR: We will compare both version 3 and version 4 fields against independent observations, as described above.
* * *
Reviewer #4:

This paper present a complete documentation of the Polar Pathfinder sea ice drift and age dataset hosted on the National Snow and Ice Data Center as well as the improvement made from Version 3 to Version 4. The main improvement in the drift dataset is in the optimal interpolation scheme used to merge the satellite, buoy and free-drift estimates into one single dataset. The new scheme now use a weighted average of different drift products, with the

weight calculated from their respective errors, and a radius of influence that is based on the decorrelation spatial scale derived from observations. The main improvement in the age dataset is the use of the new ice drift dataset discussed in this paper. Main results include a significant reduction in large spatial gradients at the junction where buoy and satellite products were used in the merged product (mainly in the earlier part of the record where buoy data is more scarce), faster sea ice drift speed and more older ice again in the earlier part of the record (when compared with Version 3).

This paper was long awaited. As stated by the authors, the NSIDC drift and age data is used by several groups, but was lacking a single source in the scientific literature that can be cited describing in details the method used to create the dataset. The paper is mostly well written except in places where some editing is required (see example below).

I recommend that the paper be published after addressing the comments below.

Comments:

Title: Mention polar pathfinder and/or NSIDC in the title.

Page 3, line 13: "...along as well as nothing the changes...". This sentence needs to be rephrased.

Page 3, line 16: Include a table listing all data sources, the time period when the data is used in the merged product and the spatial resolution of the data product.

Page 3, line 19: It should be mentioned here that the optimal interpolation scheme is described in more details below, so the reader knows there is more than just Figure 1 describing the new scheme.

Page 4, line 19: 7.23 cm/sec is still larger than the mean sea ice drift in the Beaufort Gyre. It also means that the satellite drift estimates are not continuous. Later the authors mention that temporal interpolation into a weekly product and spatial (optimal) interpolation smooths the derived velocity field. A sentence should added to tell the reader that this is discussed later on in the paper.

Page 4, line 19: The over-sampling procedure need to be described. Are the images linearly interpolated to get this sub-pixel resolution? If so, what kind of interpolation is used? I would think that the error in the over-sampled images would be a function of the interpolation scheme (linear versus non-linera), etc. Please discuss. Also, how large is the window that is translated to get at the maximum correlation?

Page 4, line 33: Please rephrase in the active form for clarity.

Page 5, line 11: "... and there was NO provenance..." instead?

Page 5, line 19: The spatial resolution of the wind product needs to be stated.

Page 6, line 3: This corresponds to a weighted average drift speed for a 36-hour period with weight 1, 2, 1 for the midnight-noon, noon-midnight, and midnight-noon (next day) drift

estimates. Or are these calculated as (2\*Drift(midnight-midnight) + 1\*Drift(Noon-Noon) ) / 3 for a true 1-day average? This should be clarified "cancel each other" instead.

Page 6, line 4: Give the bias estimates for all products quoted in this paragraph, in a similar manner as for the SSMI daily velocity component (Meier et al, 2000.

Page 7, line 14: Is the SHEBA GPS data part of the IABP buoy dataset? If so, we do not expect a large error because it is used in the polar path finder dataset.

Page 8, line 1: With fixed C values, the sum of all weights used for a given estimate will not add up to 1. An additional formula used to calculate the final (optimally interpolated) drift velocity (including the division by the sum of the weights) should be added.

Page 8, line 1: The error for each drift product (and used to calculate C) should be included in a Table.

Page 8, line 2: I am guessing that D will depend on the satellite product used to calculate the spatial cross-correlation, when in reality, the de-correlation length scale would be only location dependent. This should be clarified. Please state whether D is a constant or if it varies spatially.

Page 8, line 13: Interpolating or averaging? If interpolating, what interpolation scheme (i.e. bi-linear interpolation, kriging, spline) is used?

Page 9, line 5: This is the first time mentioned. What is the resolution of the daily product if not the same as the weekly product?

Page 9, line 6: It is desirable to use the weekly product for all applications, no? I.e. not just for trend estimates.

Page 9, line 21: Put "in year" in parenthesis at the end of sentence for better flow.

Page 10, line 9: Error in drift vectors can lead to convergence of different track into the same grid cell. I believe that the oldest of the tracks is retained in the algorithm. This is another mechanism by which younger ice is lost. It should be mentioned here.

Page 10, line 10: "to spin-up to obtain". Please rephrase.

AR: We will update the text to address these issues.

---

## Author Comment (AC2) · 3 Aug 2019

The comment was uploaded in the form of a supplement: https://www.the-cryosphere-discuss.net/tc-2019-40/tc-2019-40-AC2-supplement.pdf

---

## Author Comment (AC4) · 3 Aug 2019

The comment was uploaded in the form of a supplement:
https://www.the-cryosphere-discuss.net/tc-2019-40/tc-2019-40-AC4-supplement.pdf
* * *

---

## Author Response (AR1)

**Response to Reviewers**

**Anonymous Referee #1**

The paper provides a detailed and very well written description of the new ice motion and ice age product to be delivered by NSIDC. Importance of these products is justified by a comprehensive introduction. Changes in the production chain at all stages - from individual drift components to ice age computation - are properly documented. It is illustrated that the changes at the lower level (new optimal interpolation scheme) have impact at the higher-level products (larger extent of older ice) but predominantly in the beginning of the observation period (before 1996).

Notwithstanding the high quality of the paper, in my opinion it fails to quantitatively prove that the ice motion and age products have been actually enhanced. The only evidence that ice motion was improved is qualitative - visual comparison of drift components on figures 3 and 4. Improvement of the ice age product is also illustrated only visually - a more homogeneous ice age distribution is presented on figure 6. Given the high demand for these products a proper quantitative validation is of vital importance. A section needs to be added where the ice motion is compared with other existing independent ice motion products including, for example, AMSR2 derived drift, SAR derived drift, drift of the buoys that were excluded from the optimal interpolation, etc. Although a direct validation of ice age product is probably impossible due to absence of a similar independent product, it can be validated indirectly by comparison of multi-year ice extent with products derived from passive microwave sensors or scatterometers in March - April. It is required to include in this section the widely used product quality metrics such as RMSE, bias, Pearson correlation coefficient, etc. (and preferably both for version 3 and version 4) in order quantitatively prove the enhancement of the products and illustrate applicability in different scientific domains (trend computation, assimilation in numerical models, etc).

*Thank you for the useful comments.*

*We added a section where we validated the combined motion field via comparison with CRREL buoys. This provides an independent quantitative assessment of the combined motion field and the (minor) improvement from Version 3 to Version 4. As the reviewer notes, validation of ice age is not really feasible. We have added a paragraph further discussing validation and also discuss a recent paper (Lee et al., 2017) that included the ice age in an intercomparison with other ice age/type estimates, finding good consistency between them.*

Minor comments No grammar mistakes or typos were identified and the minor comments only concern few clarifications / corrections that are needed in the text.

P3, L23 and L25. Some authors distinguish feature-tracking (detection of individual keypoints on two images => description of keypoints by a binary vector based on =>

brute-force matching of keypoints, eg. SURG, ORB, etc) from pattern-matching (maximum cross correlation continuously applied to every n-th pixel) [e.g. Rublee et al., 2011, Berg et al., 2014, Korosov et al., 2017]. Maybe a consistent use of "patternmatching" is preferable in these two cases.

*We changed the phrasing from "feature-tracking" to "pattern-matching" in the manuscript.*

P4, L18. How was the effectiveness of 4X oversampling estimated?

*The 4X method was selected empirically by trying different values and weighing the improvement in motion accuracy versus complexity and computational expense and 4X was found to be the best choice. Text has been added to clarify this.*

P5, L5. What is the criterion for omitting rogue vectors? Difference from median of vectors in the vicinity? What is the threshold for screening?

*The spatial filtering requires at least two other neighboring grid cells to have displacements within two grid cells of each vector. The two-grid cell limit has been added to the text for clarification.*

P5, L12. What were the thresholds used in V3 and V4 for filtering PMW vectors?

*The threshold was the same in both versions, but a bug in the code was discovered that reduced the number of vectors used in the neighborhood filtering process. This effectively increased the number of vectors required for a "good" match to get a valid motion vector, and thus reduced the overall number of retrieved vectors. We've added a sentence to clarify this.*

P5, L22. 1% seem to be quite an underestimation of ice drift speed. In addition, this relation cannot be constant in space and time. With the available large amount of collocated data on wind speed and observed ice drift it should be quite simple to illustrate validity of this 'constant 1%' assumption. It would be important to justify it, e.g. in the Discussion section where the relationship between ice drift and wind speed is illustrated spatially and temporally.

*Yes, 1% is likely low. As noted in the text, in the original algorithm development, 1% was chosen based on Thorndike and Colony (1982). But we recognize that this source is likely out of date, particularly in light of observed increases in ice speed relative to wind speed (Spreen et al., 2011). However, as noted in the text, in Version 4, we did not change the wind-ice relationship, though we plan to investigate this further for a future version and will likely change it. We did run the product with a 2% relationship and found that there was very little change in the combined motion product. This is not unexpected since winds are given a low weight compared to the other sources. We have added some text here to acknowledge that 2% is likely more accurate.*

P7, L12. Is there a proof that the motion is "largely unbiased"? It is important to add a validation section (as explained in the general comment section above) to prove this statement.

*We added that the "largely unbiased" assessment comes from previous studies. And we have added a validation section of the combined motions to provide a quantitative assessment.*

P8, L5 and L6. What is the impact of values of C and D parameters on the drift speed quality (visual appearance) and accuracy (as can be retrieved from validation)? How sensitive are the motion and age products?

*The values of C and D were derived during the original development of the product and, as noted in the text, were derived empirically. C was derived via comparisons with buoys, while D was derived based on spatial cross-correlations between estimates at different separation distances. In this paper, we effectively examined the sensitivity to C and D by changing the weighting from the closest 15 vectors to the 15 highest weighted. This effectively increased the distance scale for high C observations (i.e., buoys). As described later in the paper, this did have a noticeable effect on the visual appearance of the fields (removing much of the "bulls-eye" seen in Figure 3), but did not change the motion statistics much, as shown in the statistical results in the new validation section. The sensitivity to C is also seen in the test of doubling the wind speed forcing, which had little effect on the overall combined motions because the weight of the winds is relatively smaller.*

P9, L26. I'm confused by the phrase "...all parcels in the 12.5 km ice age grid are initialized with an age-class...". Does it mean that there are several parcels per grid cell? How many?

*There can be more than one parcel in a grid cell. Each grid cell is initialized with one parcel, but as parcels advect, two (or more) parcels can merge into the same grid cell.*

P10, L19. How much "substantially"? It would be nice to have a numerical characteristics to compare V3 and V4.

*As we're describing the visual "speckling", it is a qualitative assessment, not quantitative. We've changed the text to note this and removed "substantially" since we don't quantify the speckling effect. The quantitative effect is shown in Figure 9, showing increased 4+ year old ice in Version 4.*

P11, L7. I don't quite agree that the difference between V3 and V4 is "fairly consistent over time". It grows from almost 0 (between 1980 and 1986) to almost 1 cm/s (between 2012 and 2017)! It clearly contributes to the difference in drift speed trends between v3 and v4. But which one is more correct? It is very unfortunate that proper quantitative validation is not provided. Maybe this difference is an indication of uncertainty of the motion product and the observed trends are actually statistically insignificant?

*This is a very good point. We've added to the discussion to explicitly note these changes, which are due to the passive microwave sensor changes and their different temporal and spatial sampling, which affects the discretization of the motion estimates. We've added text to further discuss this. And we have added a validation section that quantitative compares the motions from the two versions to independent buoy estimates.*

**Anonymous Referee #2**

Review of "An enhancement to sea ice motion and age products" by Mark A. Tschudi et al. TC-2019-40

General comments:
1) 7-10: Stale opening sentences in the abstract. Reads boring, repeats phrases.

*Edits have been made to the beginning of the abstract to remove repetitiveness and to flow better.*

2) Abstract: Suggest to provide more "scientific" results/summary. And drop the first few sentences.

*Quantitative results have been added to the abstract.*

3) The ms is a bit plain and could be lifted by addition of further investigation of the ice-motion and ice-age data sets and discussion of the results.

*We have added a validation section and expanded discussion of the results in various places throughout the paper.*

Specific comments:

1/12-13: Pls specify/give example on how they "are not substantially different between the versions."

*This wording has been changed in response to the above comment about the abstract. The words have been replaced by more precise and quantitative terminology.*

1/18: "recent years" or "recent decades"?

*We rewrote this as "over the 30+ year time series"*

1/26: Suggest to rephrase "it is more difficult to draw solid quantitative".

*We replaced "draw" with "determine".*

2/26: Correct "will expand greatly with the launch of the NASA ICESat-2 in September 2018" as all this is happened (i.e., it is not longer in the past).

*The text has been updated to reflect the launch of ICESat-2.*

2/27: Could mention Op IceBridge in this paragraph.

*We added a mention of IceBridge, including a reference to sea ice estimates from it.*

3/8-9: Redundant?

*Rewrote to eliminate the redundancy.*

3/12-14: Shorten.

*We removed a sentence to shorten this section.*

3/17: Change "ice motion" to "sea ice motion".

*Changed.*

3/25-26: Provide info on typical repeat frequency of "Two geolocated, spatially coincident, temporally-consecutive satellite images".

*Added sentence that the typical interval is 1 to 3 days.*

4/18: How is the oversampling rate of "4" motivated?

*Sentence added to provide information on motivation/rationale for the 4X oversampling.*

4/34: This statement is not correct as is: "AVHRR was discontinued after 2000." Please qualify or remove.

*Rewritten to clarify that the AVHRR discontinued being used in the motion product after 2000.*

5/3: How is the threshold of "0.4" motivated?

*A sentence was added to address this. Different thresholds were tried and 0.4 was found to be best in terms of not being overly stringent (removing "good" matches) while still eliminating most "bad" matches.*

5/8-15: It is not clear how exactly previous versions dealt with input PM data. Can you separate into composite versus swath or similar?

*No swath data were used. All Tbs are gridded daily composites, including the near-real-time source. We added "gridded" in this section to make that clear.*

5/19-24: The assumption that sea ice moves at 0.01 of the wind speed (for the Arctic) needs to be reviewed, especially in an environment of highly variable and increasing

wind speeds. –> Underestimate of the ice speed. I.e., Rampal et al. [2009], Positive trend in the mean speed and deformation rate of Arctic sea ice, 1979–2007, J. Geophys. Res., 114, C05013, doi:10.1029/2008JC005066.

*We added text and the Rampal reference (as well Spreen et al., 2011) to explain the use of 0.01 and to note that this is likely not optimal. We will investigate changing this, but as this paper is documenting Version 4 in which 1% is still used. We conducted an analysis to test the sensitivity to the wind speed and found the changes in the combined motion field to be small. This is not surprising since winds are weighted much less than other sources.*

5/29: Replace "data" in "These buoys monitor meteorological and oceanographic data", i.e., to read "conditions" or "states".

*Changed.*

5/33: Mention explicitly that there are too few sea-ice buoys in the Southern Ocean.

*Added.*

8/23ff: It is not clear how the few PM (or combined) motion vectors are treated to derive a broad map of sea-ice motion (on EASE grid)? It appears as if severe extrapolation is taking place.

*The motions are combined via optimal interpolation, which uses a weighted distance method, as described at the beginning of the paper. The weighting takes account of the distance and distribution of each vector within the valid distance range of the target interpolation point. Weights are also dependent on the quality of the motion source, but for the Antarctic, all motions are passive microwave. As discussed in this section, if there are too few motions, even the weighted method may not produce realistic fields, which was the case in Version 3 due to incorrectly removing "good" vectors. This was fixed in Version 4 and yields more realistic Antarctic motion fields. We've added text to better explain this.*

8/24: There are several experiments with decent buoy arrays available for some parts of the Antarctic sea-ice zone. Why not use some of those to at least assess the skill of the product... and to possibly explore the suitability of Antarctic ice-buoy data to provide information into the ice-motion product discussed here.

*Yes, there are some Antarctic buoys, such as the IPAB buoys. But the coverage in space and time is much less than in the Arctic. We may consider adding these and other available buoys to the Antarctic fields at some point. Regarding validation, we've chosen not to so for now. We plan to look at these as part of a more comprehensive enhancement for a future major version update when resources allow. The focus of the product has been the Arctic, and age is only produced for the Arctic.*

9/2: The netCDF file should include an additional mask (0/1) where one can mask all gridded ice motion that is "too far" from an actual observation, where the value of "too far" needs to be discussed.

*We agree that this would be a good idea and will be considered in the next version of the product. We do note that a flag value is included in the error estimate in the daily combined fields to indicate where all vectors are far from the interpolated grid cell. A sentence has been added to note this.*

9/32: How is the limit of "16 years" for the maximum ice age set? Physical motivation?

*This was designed in the original implementation under the rationale that there is little or no ice of this age.  Most of the small fraction of 16-year ice has other ice ages associated with that grid cell.  For these cells, the algorithm reassigns the ice age to the oldest age of the remaining ice in the cell that is less than 16 years. Eliminating the tracking of parcels older than 16 years also increases computational efficiency.  Note that all ice older than 5 years is treated as one group in the ice age browse imagery, and in analyses we perform, such as the annual NOAA Arctic Report Card.*

10/19: There is not quantitative measure of how V4 ice age as improved relative to V3: "there is less "speckling"".

*The quantitative assessment is in the change in the proportion of older ice versus younger ice discussed in Section 4 and shown in Figures 9 and 10. We've added text to explicitly link these sections.*

10/21ff: In discussing the relative "ageing" of Arctic ice from V3 to V4 there are no physical details provided as to what process would be the main driver of this change.

*As noted earlier, the issue in Version 3 is the discontinuity in the buoy estimates and other source motions due to sub-optimal spatial weighting, as found by Szanyi et al. (2016). The discontinuity is largely removed with the improved weighting. We added text here to more clearly describe this.*

11/1ff: The discussion of trends and variability in ice motion & age between V3 and V4 should be more quantitative. – Also, regional contributions should be explored.

*We've added more quantitative discussion on the motion and age products. The main focus of the paper is to present the new version of the products, compare and validate their differences, and briefly show long-term trends as an example of their application. While regional variability is certainly of interest and worth investigating, we feel that it is beyond the scope of this paper.*

12/7: Correct "Fennoscandian peninsula." to "Fennoscandian Peninsula." (upper case)

*Corrected.*

Fig.7: There seems to be a cyclical signal in the ice-speed difference between V3 and V4. Decadal or perhaps 11 - 12 years. Can different PM sensors be the reason for this? Or the speed magnitude??

*Yes, the change in sensors had an effect on the magnitude due to differing temporal or spatial sampling. We've added text at the beginning of Section 4 to discuss this further.*

Fig.9: The version difference in ice age for 4yr+ is not well explained.

*We've added more to the discussion of Figure 9.*

Fig.10 & 11 are not well explained/discussed.

*We have added more discussion of both figures.*

**Anonymous Referee #3**

Review of "An enhancement to sea ice motion and age products" by Tschudi, M., et al.

Summary: This contribution attempts to illustrate the enhancements - to be understood mainly as extension - of two sea-ice products issued by the National Snow and Ice Data Center (NSIDC), namely the NSIDC sea-ice motion data set and the NSIDC seaice age data set. The latter is based on the former. The manuscript advertizes the data sets, informs a bit about the history of these two data sets and describes briefly changes made to the processing which potentially led to an enhancement in quality of both products.

My overall impression is that this paper is not suitable for publication and should be rejected.

It lacks essential information about the retrieval procedure, and the retrieval uncertainties. It further lacks results of an evaluation. It presents trends which seem artificial. It is incomplete in terms of geographical coverage. It contains errors. There are many open technical questions which are not answered in the manuscript and also not in the respective documentation of the data set(s) on the NSIDC web pages. This paper is written as if it extends a reference benchmark paper where all the required missing details could be found. But this is not the case. Such a benchmark paper does not yet exist for the sea-ice motion product. As the authors stress, the sea-ice motion data set is unique, it has a unique length, and it allows unique applications. And therefore it requires a unique extensive high-quality paper first, in which the reader and the data users can learn about all details and limitations associated with the data set and its generation and evaluation.

General Concerns:

GC1: No systematic evaluation of the products has been undertaken - neither for version 3 nor for version 4 of the sea-ice motion product. Also the associated newest sea-ice age data set is not evaluated. In your case, it is not sufficient to just compare version 3 and version 4 of the product because a systematic, detailed evaluation of version 3 products is missing in the scientific literature. There is hence no benchmark against which this new version 4 can be quantitatively referenced. Section 2.2 does not provide new results. There is no indication of a useful sea-ice motion retrieval uncertainty provided along with the product, like is done for sea-ice concentration and thickness data sets. The authors do not present results of an evaluation neither of the newly derived components of the sea-ice motion entering the gridded product nor of the gridded product.

*There has been significant validation done on the basic algorithm, particularly the MCC approach (Kwok et al., 1998; Kwok, 2008; Meier et al., 2000), as well as on a previous version of the specific product (Sumata et al., 2014; Sumata et al., 2015).*

*Uncertainty estimates are included in the daily combined motions, based on the optimal interpolation and the relative weights of the source data. We've added mention of this to the text.*

*Sea ice age is difficult to directly validate as there is not validation with sufficient accuracy and coverage. We have added a reference (Lee et al., 2017) to an intercomparison study that shows good overall agreement between the ice age data and other ice type/age products.*

GC2: The reader and data set user is informed about user statistics, the importance of the two data sets, some selected bits of the history of the retrievals, and a relatively unspecific description of the changes made to the methods which leaves many open questions. This is, however, potentially not what a reader of this paper and user of this data set would have expected for the following reason: There is no specific paper in which the various retrieval processes, their uncertainties, the caveats of the different spatio-temporal resolution of the input data sets, a detailed description of the merging (optimal interpolation) approach and its uncertainties have been published so that the full package of detailed, high-quality information is visible at a glance. The retrieval, the input data, the pre-processing steps all these are not transparently described. In other words: A benchmark reference paper containing all bits and pieces is missing so far. And in this context this paper about an "enhancement" seems of doubtful value.

*Such information is provided is the User Guide for the product, provided by NSIDC. We chose not to include this information in the manuscript in the interest of brevity. We've added reference to the NSIDC User Guide. We agree that a peer-reviewed document on the original development of the product would have been useful. However, the original product developer chose not to submit such a paper. As such, our purpose with this manuscript is not to try to recreate a history of which we do not know all of the details, but to document the changes and improvements the current team has made to the newest product as well as giving an overall summary of the processing that is described in the NSIDC User Guide.*

GC3: The introduction is a nice compilation of recent work dedicated to changes in Arctic sea-ice area / extent, multiyear ice fraction and thickness and in Antarctic sea-ice area and extent. But: during the past two decades or so various other approaches for sea-ice motion retrieval have been developed and the respective data sets are also in use. This paper lacks a review of this work. The retrieval method is not put into context of the current research landscape in this field. This applies to new algorithm developments (both method and input satellite data) as well as evaluation studies. What I, in this context, understand the least, is that despite evidence exists in the literature from various groups using predecessors of the ice motion data set (mainly version 2 and 3), that the inter-sensor inconsistencies cause artificial trends computed from the ice motion product and render parts of the product not useful, you do not comment about this.

*We've added references to other ice motion products in the Introduction section. We apologize for this oversight.*

GC4: I also miss an evaluation of the ice-age data set and/or more quantitative statements about its reliability and potential uncertainty. I do not rate a comparison to the previous version of the data set as providing enough evidence for a proven enhancement. Such a comparison provides only qualitative information about the potential sign of the enhancement. Any quantitative information which would go beyond the comparison to the previous version is lacking. There is a list of specific comments which I can provide on request if need be.

*Quantitative assessment of the ice age product is difficult because there are no high-quality validation data sets to compare to. Since age is directly derived from the motion product, improvements in motion estimates should carry on to the age product. We also show qualitatively that the spatial distribution of age is more realistic. Finally, we have added a reference that compares the ice age product to other ice type/age products and demonstrates that our product is reasonably consistent with the other estimates.*

**Reviewer #4**

This paper presents a complete documentation of the Polar Pathfinder sea ice drift and age dataset hosted on the National Snow and Ice Data Center as well as the improvement made from Version 3 to Version 4. The main improvement in the drift dataset is in the optimal interpolation scheme used to merge the satellite, buoy and free drift estimates into one single dataset. The new scheme now uses a weighted average of different drift products, with the weight calculated from their respective errors, and a radius of influence that is based on the decorrelation spatial scale derived from observations. The main improvement in the age dataset is the use of the new ice drift dataset discussed in this paper. Main results include a significant reduction in large spatial gradients at the junction where buoy and satellite products were used in the merged product (mainly in the earlier part of the record where buoy data is more scarce), faster sea ice drift speed and more older ice again in the earlier part of the record (when compared with Version 3).

This paper was long awaited. As stated by the authors, the NSIDC drift and age data is used by several groups, but was lacking a single source in the scientific literature that can be cited describing in details the method used to create the dataset. The paper is mostly well written except in places where some editing is required (see example below). I recommend that the paper be published after addressing the comments below.

*Thank you for the useful comments.*

Comments:
Title: Mention polar pathfinder and/or NSIDC in the title.

*Added NSIDC to the title.*

Page 3, line 13: "…along as well as nothing the changes…". This sentence needs to be rephrased.

*Rephrased.*

Page 3, line 16: Include a table listing all data sources, the time period when the data is used in the merged product and the spatial resolution of the data product.

*We've added a table, Table 2.*

Page 3, line 19: It should be mentioned here that the optimal interpolation scheme is described in more details below, so the reader knows there is more than just Figure 1 describing the new scheme.

*Added.*

Page 4, line 19: 7.23 cm/sec is still larger than the mean sea ice drift in the Beaufort Gyre. It also means that the satellite drift estimates are not continuous. Later the authors mention that temporal interpolation into a weekly product and spatial (optimal) interpolation smooths the derived velocity field. A sentence should added to tell the reader that this is discussed later on in the paper.

*A sentence has been added.*

Page 4, line 19: The over-sampling procedure need to be described. Are the images linearly interpolated to get this sub-pixel resolution? If so, what kind of interpolation is used? I would think that the error in the over-sampled images would be a function of the interpolation scheme (linear versus non-linear), etc. Please discuss. Also, how large is the window that is translated to get at the maximum correlation?

*Further details on the oversampling are provided and the window size is noted.*

Page 4, line 33: Please rephrase in the active form for clarity.

*This is addressed in response to a comment from another reviewer.*

Page 5, line 11: "… and there was NO provenance…" instead?

*Changed.*

Page 5, line 19: The spatial resolution of the wind product needs to be stated.

*The spatial resolution was added and reference made to the added table.*

Page 6, line 3: This corresponds to a weighted average drift speed for a 36-hour period with weight 1, 2, 1 for the midnight-noon, noon-midnight, and midnight-noon (next day) drift estimates. Or are these calculated as (2*Drift(midnight-midnight) + 1*Drift(Noon-Noon) ) / 3 for a true 1-day average? This should be clarified

*There are two observations, both encompassing 24 hours: midnight to midnight, and noon to noon. These are averaged to get one 24-hour estimate. However, you are correct, the four observations combined span 36 hours.*

Page 6, line 33: Write "cancel each other" instead.

*Changed.*

Page 6, line 4: Give the bias estimates for all products quoted in this paragraph, in a similar manner as for the SSMI daily velocity component (Meier et al, 2000.

*We've added bias estimates based on Meier et al., 2000 and similar studies.*

Page 7, line 14: Is the SHEBA GPS data part of the IABP buoy dataset? If so, we do not expect a large error because it is used in the polar path finder dataset.

*The SHEBA data are not part of the IABP dataset.*

Page 8, line 1: With fixed C values, the sum of all weights used for a given estimate will not add up to 1. An additional formula used to calculate the final (optimally interpolated) drift velocity (including the division by the sum of the weights) should be added.

*We added a note to indicate that the weights are normalized so that they sum to one.*

Page 8, line 1: The error for each drift product (and used to calculate C) should be included in a Table.

*These values were calculated early in the development of the original product. There is not a table of errors for each drift product. Errors are discussed for each product in the NSIDC user guide for the motion product: https://nsidc.org/data/nsidc-0116.*

Page 8, line 2: I am guessing that D will depend on the satellite product used to calculate the spatial cross-correlation, when in reality, the de-correlation length scale would be only location dependent. This should be clarified. Please state whether D is a constant or if it varies spatially.

D *may depend to some degree on the satellite product, but it's really the correlation length scale of the motions themselves, which is source independent. In any event, it is treated as a constant in the product processing.*

Page 8, line 13: Interpolating or averaging? If interpolating, what interpolation scheme (i.e. bi-linear interpolation, kriging, spline) is used?

*It is optimally interpolating, referring back to the method noted in the previous sentence. We added "optimally" and "kriging" for clarity.*

Page 9, line 5: This is the first time mentioned. What is the resolution of the daily product if not the same as the weekly product?

*The daily and weekly motions have the same spatial resolution (25 km). The weekly average smooths out day-to-day variability due to the coarseness of the resolution. We added some text for clarity.*

Page 9, line 6: It is desirable to use the weekly product for all applications, no? I.e. not just for trend estimates.

*Yes. We've changed the text to note this.*

Page 9, line 21: Put "in year" in parenthesis at the end of sentence for better flow.

*Edited to flow better.*

Page 10, line 9: Error in drift vectors can lead to convergence of different track into the same grid cell. I believe that the oldest of the tracks is retained in the algorithm. This is another mechanism by which younger ice is lost. It should be mentioned here.

*We've added text in the previous paragraph to note this.*

Page 10, line 10: "to spin-up to obtain". Please rephrase.

*Rephrased for clarity.*

---

## Author Response (AR2)

*We wish to thank the reviewers for their very helpful comments. In particular, we thank Reviewer #2 who has tirelessly went through the paper in fine detail and offered many good suggestions, both on improvements to the manuscript and for future improvements to the products.*

*The most significant changes we made are (1) adding validation of the source motion estimates via comparison with the buoy estimates, and (2) adding further details to the CRREL buoy comparison. In the interest of keeping the main manuscript focused and reasonably concise, we have added a supplement that contains this material. We also moved the rest of the CRREL buoy analysis (Section 2.4 in the previous draft) into the supplement. This keeps that material in one place, which we feel will make it easier for readers to follow both the main text and the supplement material.*

*We have responded to all of the comments noted by the reviewers and have made all the changes that are feasible to address the reviewers' concerns. Our responses to each comment are inline below.*

**REVIEWER #1**
The authors have significantly improved the manuscript, added validation results and evaluation of uncertainties. It would be nice to address only two minor comments.

1. The seasonal variability in the difference between V3 and V4 is attributed to the higher ice drift speeds in summer (on page 19, line 3). Does that mean that the difference between the products is rather multiplicative than additive? Please illustrate that by an extra plot showing relative difference (V4/V3) in addition to the absolute difference (V4 - V3, on Fig. 7). If the factor is nearly constant in various seasons, then it is indeed mostly due to changes in temporal sampling. Otherwise additional explanations are needed.

*We plotted V4/V3 (see below) and seasonality is still evident. So, this means that the difference is not strictly multiplicative. However, the effect is likely still related to seasonality. An illustrative example, V3 has winter motions of 1 cm/s and summer motions of 3 cm/s. If it is multiplicative, then if V4 winter motion is 2 cm/s, the summer motion would be 6 cm/s – i.e., the ratio of V4/V3 would be 2 for both summer and winter while the difference would be 1 cm/s in winter and 5 cm/s in summer. But, if instead V4 summer motion was 9 cm/s, then the summer ratio would be 3 instead of two. The difference would be 1 cm/s in winter and 8 cm/s in summer.*

*Thus, we feel it is still primarily a seasonal effect of higher speeds during summer, but it is not strictly multiplicative. Another factor that we didn't originally include is the fact that there are fewer passive microwave motions during summer because surface melt limits correlation between images. We've added some text to discuss this more, including the lower number of passive microwave vectors during summer.*

[Figure]

2. The statement in Conclusions (page 21, line 16) "these artifacts did not substantially affect the weekly sea ice motion or age fields" seem to contradict the statement in Section 4 (on page 19, line 12) "... largest effect of the version change for ice age is ... the amount of multi-year ice in the early part of the record...". Consider rewriting conclusions.

*We edited the Conclusion to be consistent with Section 4.*

**REVIEWER #2**
I am referring to my first review of this manuscript for the summary of the paper.

I begin with a reply to a few points from the rebuttal letter to my first review. Obviously, you do not need to comment to these but are of course invited to do so.

General Concerns from 1st review:
GC1: No systematic evaluation of the products has been undertaken - neither for version 3 nor for version 4 of the sea-ice motion product. Also, the associated newest sea-ice age data set is not evaluated. In your case, it is not sufficient to just compare version 3 and version 4 of the product because a systematic, detailed evaluation of version 3 products is missing in the scientific literature. There is hence no benchmark against which this new version 4 can be quantitatively referenced.

Section 2.2 does not provide new results. There is no indication of a true sea-ice motion retrieval uncertainty provided along with the product, like is done for sea-ice concentration and thickness data sets. You do not present results of an evaluation of the newly derived components of the sea-ice motion entering the gridded product as well as the gridded product.

Reply by the authors:
A) There has been significant validation done on the basic algorithm, particularly the MCC

approach (Kwok et al., 1998; Kwok, 2008; Meier et al., 2000), as well as on a previous version of the specific product (Sumata et al., 2014; Sumata et al., 2015).

B) Uncertainty estimates are included in the daily combined motions, based on the optimal interpolation and the relative weights of the source data. We've added mention of this to the text.

C) Sea ice age is difficult to directly validate as there is not validation with sufficient accuracy and coverage. We have added a reference (Lee et al., 2017) to an intercomparison study that shows good overall agreement between the ice age data and other ice type/age products.

My comment to the authors' reply:

To A): I can agree to that. However, this is version 4 and you compare version 4 to version 3. Version 3 has not been assessed even though there have been substantial changes between version 2 and version 3, e.g. in the selection of which passive microwave data are used (see Haumann et al., 2016). Particularly when it comes to the long-term stability and the consistency across different satellite sensor products used - which is a mandatory element of product maturity for such a long-term and important data set as the one you are presenting here, it seems sub-optimal to refer to evaluation activities which only target the basic algorithms. One could have expected a considerable move forward here. I therefore appreciate that you now carried out an evaluation of the combined Arctic product with CRREL ice-mass balance buoy data. This is a good start.

*We retitled Section 2.2 to be clear that the discussion is a review of general uncertainties from previous studies.*

To B) I am aware of these uncertainty estimates. These are very much based on the merging process of the data. The mentioned relative weights are only valid / can only be computed where there is a buoy for comparison - plus some correlation length scale around these - leaving the majority of the investigated region void of (detailed / accurate) quality information; this applies in particular to the Antarctic, doesn't it?

What about the uncertainties of the individual products, i.e. the AVHRR ice motion vectors or the AMSR-E ice motion vectors?

To reviewer #4, who stated: "Page 8, line 1: The error for each drift product (and used to calculate C) should be included in a Table.", you replied: "These values were calculated early in the development of the original product. There is not a table of errors for each drift product." "early in the development of the original product" is exactly what I am referring to. It appears there was very limited effort to re-evaluate the newer original data sets input into the merged product. If we'd compare this to the situation in the sea-ice concentration community then this would mean that this community would try to sell a new improved sea-ice concentration data set - referring to the accuracy of the SMMR sea-ice concentration data sets. --> See my general comment GC1 to the revised manuscript.

*In response to your suggestion, we have added a supplement to provide a brief analysis of the source data products. We compared the source motion estimates with buoys for selected periods. That is all that is feasible at this time. We do fully agree that a complete reassessment and uncertainty evaluation would be very useful. We hope to do that, pending resources. We hope*

*that the short evaluation that we've added will suffice for now. Our results agree generally well with other comparisons discussed in Section 2.2*

To C) Certainly I agree that it is challenging to get a reasonable set of evaluation data for this purpose. However, I mean, your group has been THE group to build and maintain this data set for over a decade now and it has received great attention. To my opinion it would have been more than timely to come up with some attempts and advanced ideas to evaluate this data set - at the latest after the publication of Korosov et al. (2018). Therefore, I am inclined to rate the efforts carried out into this direction as not sufficient. While the reference added (Lee et al., 2017) mitigates this impression a bit, it cannot replace a decent evaluation and/or inter-comparison study which underlines the improvement of version 4 over version 3 and underpins the credibility of this data set.

*We appreciate the reviewer's assessment that we have been "THE group" to build and maintain this data set. The reviewer's suggestions to improve are greatly appreciated. They are on our list to do for a future version when we have the resources to do so. This paper focused on documenting Version 4 and, to the extent possible, documenting the basic methodology of the previous version. A primary motivation was to address the motion artifact identified by Szanyi and correct other known issues. Version 4 is not meant to be a comprehensive reassessment of the product because there have not been resources to do so.*

GC2: The reader and data set user is informed about user statistics, the importance of the two data sets, some selected bits of the history of the retrievals, and a relatively unspecific description of the changes made to the methods. This is, however, potentially not what a reader of this paper and user of this data set would have expected for the following reason: There is no specific paper in which the various retrieval processes, their uncertainties, the caveats of the different spatio-temporal resolution of the input data sets, a detailed description of the merging (optimal interpolation) approach and its uncertainties have been published so that the full package of detailed, high-quality information is visible at a glance. The retrieval, the input data, the pre-processing steps all these are not transparently described. In other words: A benchmark reference paper containing all bits and pieces is missing so far.

Reply by the authors:
Such information is provided is the User Guide for the product, provided by NSIDC. We chose not to include this information in the manuscript in the interest of brevity. We've added reference to the NSIDC User Guide. We agree that a peer-reviewed document on the original development of the product would have been useful. However, the original product developer chose not to submit such a paper. As such, our purpose with this manuscript is not to try to recreate a history of which we do not know all of the details, but to document the changes and improvements the current team has made to the newest product as well as giving an overall summary of the processing that is described in the NSIDC User Guide.

My comment to the authors' reply:
I do understand the challenge behind trying to unwrap the bits and pieces of the original (versions 1 and 2) processing of this data set. I would have hoped that you would be able to undertake this tedious work - because nobody else can do it; you are closest to those people and

institutions which developed the data set. I see that with my review I wasn't able to convince you that for a journal such as "The Cryosphere" it pays back to double efforts to accept this challenge. I assume that this has been a question of funding and therefore need to leave this issue where it is. It is a pity. As I stated in my original review (actually you never received the long version which made me to reject the previous version of your manuscript) it is very tedious for a data user to climb down into all the web pages that are linked from the NSIDC User Guide and it is in addition relatively frustrating that the information one searches for is partly quite limited or simply not available.

Please also see my GC1 to the revised manuscript in this context.

*As stated above, we agree with the reviewer that this work should be done and we do hope to do it in the future when/if we procure funding. Under current funding, we were somewhat limited on what we could accomplish. The goal of this paper was to document the changes for V4, address the issue noted by Szanyi et al. (2016), and since the product had never been documented in a peer-reviewed article, we wish to provide at least some more background than what has been previously published.*

####################

Now comes the review of the revised version of the manuscript

Following up with the authors' reply to my general concerns formulated in my review of the first version of this manuscript, and with the comments of the other reviewers, I am convinced that the manuscript will become acceptable after major revisions.

These revisions should cover the general comments GC1 through GC3 (see below).

While the remaining two general comments (GC4 and GC5) are just ideas to think about, I ask you to consider the specific comments. Some of these - to my opinion - require attention for an adequate understanding of your paper and for having it placed in "The Cryosphere". By following the suggestion I formulate in GC1 you might be able to solve some of the specific comments.

Finally, I found a good number of typos and formulations which require attention which I listed separately in the last part of my review; here you will also find suggestions related to the figures.

General comments:
**GC1** refers to the insufficient description of production / evaluation of the "old" basic retrieval of the ice motion. I can understand that it might be too time consuming to dig into all the old documents, web pages, reports, folders, whatsoever left behind by the scientist team responsible for the creation of versions 1 through 3 of the ice motion product. I understand that you don't want to do it and that you don't receive funding for it. Still, this information is mandatory for a reader and/or user to fully understand what this product is about, what its limitations are, and where substantial potential for improvement exists. This would be of high value for other scientists to take over and to develop alternative products.

Since you don't want to write a "benchmark paper" as I suggested in my previous review [even though that one would be good for The Cryosphere, while this manuscript here might be more suitable for Earth System Science Data], I suggest that you provide list of the unknowns, open parameters or assumptions, and original evaluation activities in form of a few tables and/or illustrations put into an appendix to this paper.
Example 1: You cannot find out how C and D in Eq. 1 were computed exactly? What the values are? What the correlation lengths are? Mention this explicitly in one of the tables.

*We did not think that a table would adequately address the reviewer's concerns. Instead, we have added text where appropriate to provide more details on the original processing and further clarify what the unknowns are.*

*We have added further information on the C and D values in Equation 1. And we note that the original derivation is not known and that values may not be optimal for the data quality.*

Example 2: You cannot find out whether the 4X over-sampling is carried out for all 37 GHz channel data of all three instruments SMMR/SSMI/SSMIS? Mention this explicitly in one of the tables.

*The 4X oversampling is done for satellite estimates and is noted in the text.*

Example 3: Nobody has yet investigated quantitatively the impact of the switching between 37 GHz and near-90 GHz data for SSMI, AMSR-E, SSMIS data on the obtained ice motion? Fine. Mention this in one of the tables (this is something noted by Haumann et al. 2016 for version 2 of the product, creating a big mess in their search for a credible time series of the ice motion for the Antarctic).

*We have added mention of this in a couple of places. First, in Section 2.1 on the gridded satellite motions, we have added a paragraph that emphasizes the differences in the satellite sources and their potential impact on the retrieved motions. We also added to Section 2.3 on the interpolation that the selection of the C values is sub-optimal. And we added a Supplement with a short evaluation of the uncertainty of the source motions where we discuss their differences and how those differences point to the impact of switching between different sources and how the product would be improved with improved weighting.*

Example 4: You cannot find out whether the input PMW ice motion vectors have been evaluated after introducing the 4X oversampling? Mention this in one of the tables.

*4X is done for all satellite sources, as noted on Page 6, line 20 of the original revision. The 4X resampling has been since the beginning of the product and was not changed for Version 4 (or Versions 2 and 3).*

Example 5: The attempts to evaluate ice motions in the Antarctic are unknown / limited to the following (...) studies? Mention this in one of the tables.

*We have added a paragraph at the end of Section 2.2 to specifically discuss the limited knowledge of Antarctic motions and the lack of evaluation/validation.*

Examples 6: What are the evaluation studies specifically targeting version 3 of the ice motion product? List the references / or NSIDC reports.

*The changes from Version 2 to Version 3 were minor, as noted in Table 1. The most significant change was the removal of erroneous buoys estimates and some poor-quality AVHRR motion fields. Correcting these fields was clearly an improvement to the product. The two other changes - using the NSIDC ice mask and the GDAL libraries – improved provenance and conformed to modern standards, but had almost no effect on the output fields. After comparisons between Version 2 and Version 3 showed that the effect of these changes was very small and further documentation of differences was deemed to not be necessary.*

I strongly recommend to be as transparent as possible and take over the responsibility to document the status of this unique long-term ice motion product as best as possible.

*We agree. This paper is an attempt toward being more transparent. It's not perfect because of the limitations in knowledge of original processing, but we feel it is an improvement in transparency. And as stated earlier, the main focus here is for the first time have a reference that documents to the best of our ability the current state of the product.*

**GC2 is referring to section 2**: When you write about "velocity components" in sections 2.1 and 2.2: do you refer to the meteorological convention, i.e. component u being defined positive from west to east and component v being defined positive from south to north? I am asking because I noted that versions 3, 4 and 4.1 of the ice motion product contain u- and v-velocity components in the direction of the EASE grid used, i.e. a positive u-component at 0degEast is a true west-to-east ice motion while at 180degEast it is an ice motion into the opposite direction. You might want to clarify this at the beginning of your paper.
I note that in section 2.3 you note that the motion is relative to the EASE grid for the gridded product.

*Yes, all vectors are retrieved and provided as u and v velocity components based on the EASE grid. We've added a sentence in Section 2.1 to make this clear.*

I note further that in section 2.4. it is not clear whether you transferred the CRREL buoy ice motion to the EASE grid notation of u and v. By the same token, users being familiar with the meteorological notion of motion might have difficulties to translate your uncertainties into their understanding of how air, water and ice movement is described, i.e. positive west --> east; positive south --> north.

*The CRREL buoys were put on the EASE grid for the comparisons and they were done on the u- and v-components relative to the EASE grid. We've added a sentence in Section 2.4 to clarify this.*

**GC3:** Information about Antarctic sea-ice motion is VERY limited. You could at least have

added similar plots like Fig. 5, Fig. 7 and Fig. 8 to illustrate that you also provide ice motion for the Southern Ocean and how this looks like in comparison to the Arctic.

Yes, it will be a challenge to discuss the jumps in the time-series between the different satellite data sources. To my opinion, because your paper is about a bi-polar (or global) data set you have to show these and you have to discuss these and you have to clearly point out the limitations to the users. I rate this a mandatory element which is still missing in the revised version of the manuscript. Alternative: You remove ALL information about the Antarctic.

*Since the Antarctic motions are part of the product, we feel it is important to include them in the manuscript. But we agree that in the previous version of the manuscript, we gave them short shrift. We have added Antarctic images for Figures 5, 7, and 8. We have also added a paragraph at the end of Section 2.2 noting that most (all?) motion validation has been done in the Arctic because of the lack of buoys in the Antarctic. We note that the Antarctic motions are of lower confidence level and users should be aware of these caveats when using the product.*

**GC4:** What would you say is a typical uncertainty estimate for the ice age (applies to Figs. 9-11 and their interpretation)? Would it be fair to say that ice-age fractions can be determined as accurate as 50 000 sqkm? ... or rather 100 000 sqkm? I would find such an estimate very useful in the context of the interpretation whether recent changes in ice age fraction are significant or within the noise created by the method.

*We agree that this is an interesting question and certainly one worth further consideration. As the reviewer is likely aware, it is difficult to find a quantitative validation source of "truth" for comparison. One can compare the age product with other ice age/type fields, at least for FYI/MYI discrimination. But we feel this is beyond the scope of this paper. It is something we hope to look into in the future.*

**GC5:** This comment applies to Page 20, Lines 13-25. In the context of the various data sources with different resolutions, different spatiotemporal coverage, and different relative weights used for the combined ice-motion product which is subsequently used for the derivation of the ice age: How much, to your opinion, do these differences have an impact on the obtained trends in ice age fractions?

*This is an interesting question as well. Our guess is that the effect is likely small. While the daily motion error characteristics of the individual various sources can be quite different, the spatial averaging from the optimal interpolation smooths out these differences. Certainly, different weighting will have some effect, but the number and proximity of observations are also key. Then, using a weekly average motion to derive age ameliorates the different levels of variability seen in daily motions. However, we don't doubt that sources have some effect. One significant change in Version 4 is the weighting scheme for combining the sources (15 highest weighed vs. 15 closest) and the effect on the daily motion fields is quite apparent in terms of the fall-off of the buoy influence (Figure 3). But the effect on the overall trend in ice age is not that large. So, this indicates that the age field and changes over the long-term are relatively insensitive to the specific combination of sources and their weightings. It would be interesting to look at age trends from motion fields derived from a single source (e.g., passive microwave vs. winds) and*

*see how the resulting age field differs. This is beyond the scope of this paper, but it is something that we would like to look into further. We thank the reviewer this idea.*

+++++++++++++++++++++++++++++++++++++++++++++++++++++++++++++++++++++++

Specific comments:
Page 3, lines 1-6:
- I am wondering whether there isn't knowledge about the relationship between ice age and thickness from earlier studies, i.e. before the NSIDC sea-ice age product era. I am sure there is. What about the book by Untersteiner for instance? Please add a respective reference to the citation of your own work.

*Indeed there is! The Maykut chapter in the Untersteiner book shows this, as do papers by Tucker et al., 2001 and Yu et al., 2004. We have added these references.*

Just a comment: It might be true that sea-ice age provides additional information - compared to maps of the multiyear ice cover - about when in a certain region sea ice will melt out during summer. But has this been proven yet in the 10+ years the ice age data set has been out? One could imagine that an accurate first-year ice versus multiyear ice discrimination is sufficient here and that it does not really play a role whether the sea ice is 2, 3, 5 or more years old when it comes to melting. Isn't the main interesting information given by the sea-ice age data set that we have a measure of how long sea ice once formed survives as a function of formation location and ice movement?

*This is a good point. As MYI ages, the thickness increases less each year, so there is less and less difference. We added some text in this section to note this.*

Page 3, paragraph starting in Line 22: This is a good start.
- Please add the work of Sumata et al. (2014, 2015) here because it gives a good overview about existing products and how these compare.

*Added.*

- Please add the work of Kwok (2008) about summer-time sea-ice motion estimation using near 19 GHz data. It is important for a reader / data user to know that there are alternatives to using NCEP winds based drift estimates during summer.

*We added Kwok (2008) to Section 2.2 where we feel it fits better because that is where we specifically discuss summer motion retrievals.*

- Please add a few sentences about the various attempts to use SAR data for deriving sea-ice motion. This is important for a reader / data user who is interested in small-scale solutions. It is further important for you yourself as this would underpin how valuable your data set is in terms of spatiotemporal coverage.

*We have added couple sentences in the introduction section on SAR along with references.*

Page 6, Lines 16-18:
It appears sub-optimal to first state that with 25 km grid resolution one can only estimate the velocity field to the nearest 25 km / day for each motion component (see lines 14-15) and then here state your product obtains useful daily motions by using the described over-sampling. The point I wish to make here is that the native resolution of the 37 GHz channels of SMMR, SSM/I and SSMIS, as given by the footprint sizes, is coarser than the grid resolution used and, in addition, these footprints are only a good approximation of the reality because of the antenna sidelobes. So what you sell to the user here is: brightness temperature observed at 37 GHz, gridded as a daily mean value into a 25 km grid, show features distinct enough that you can move over these 25 km grid cells with increments of 6.25 km. I am wondering how you assessed the improvement in accuracy stated on Page 6, Line 19. Is there a chance to include the respective results in this paper, e.g. as supplementary material / appendix? Possibly the improvement in accuracy is a function of the frequency because the native resolution is much finer for the near-90 GHz channels of SSM/I and SSMIS.

*We have added that the effective resolution (i.e., the sensor footprint) is coarser than the gridded resolution. In terms of the effect of oversampling, as noted in the text, during initial development of the product many years ago, 4X was chosen based on improved performance relative to computational expense. Since then 4X has always been used for all satellite-derived estimates. While it would be interesting to compare motions with other oversampling it is beyond the scope of this paper. The 4X oversampling is effectively a part of the basic algorithm and is thus presented as-is.*

- Another over-arching question to this oversampling: For brightness temperatures of the near 37 GHz channels of SMMR and SSM/I and SSMIS you apply 4X oversampling. How about for the 12.5 km grid resolution near-90 GHz data of SSM/I and SSMIS? And: How about over-sampling and its application to AMSR-E data at 12.5 km resolution? It appears that the description of the over-sampling procedure is not yet complete; other scientists willing to repeat your steps would not be able to do so because of a lack of information.

*As noted in our responses above, the 4X oversampling is applied to all satellite sources, as was noted in the manuscript text.*

- How does this over-sampling method compare to the continuous MCC suggested by Lavergne et al. (2010)? It is worth commenting and discussing this issue in this paper because the over-sampling seems to be something introduced relatively recently (?); it deserves to be discussed in the light of alternative choices.

*We have added a note about CMCC as an alternative approach presented in Lavergne et al. (2010).*

Page 7, Lines 16-23:
- Can you comment on minimum correlation values used by other products? How about the Girard-Ardhuin and Ezraty (2012) products?

*We added in the values used by Girard-Ardhuin and Ezraty (2012), Kwok et al. (1998), and Lavergne et al. (2010).*

- "Various thresholds ... original development of the product ... qualitatively determined ..." --> This is a relatively vague information and leads to many questions: Was this same threshold applied to all satellite data sets described so far? Has the choice of this threshold been revisited in the meantime? Based on which data set(s) this threshold was selected (when?) by the mentioned qualitative determination? Is there a paper or two to which you can simply refer to which illustrate this determination?

*Yes, it was applied to all of the satellite-derived products; we added text to make this clear. No, the threshold has not been revisited. It was determined early in the development of the MCC method and has not been changed in the product. We note that the choice is subjective, but that it is within the range of other MCC and related implementations. We added two early references that discuss the selection of the 0.4 value.*

- The description of the neighborhood filter should be precise. Neither is it clear how this is done technically (and at which grid resolution). Nor is quantified what is meant by "similar" ... direction? magnitude? How large a difference can be before the vector is considered to be spurious?

*We added more detail here to specify the quantification of the required similarity between vectors.*

Page 8: Lines 1-2: "Another change" --> What kind of a change? Please be more specific. If I understood you correctly then this is a version 4 change - so there is no excuse to not detail what apparently has been done recently by you (and not by the other scientists which developed version 1-2 (3)) of the product. Please note: Without referring to figure 4 the information about the apparent improvement by this post-processing step is completely hanging in the air.

*We've added details to explain the change in filtering between Versions 3 and 4. We also added some text to Section 2.3 on the effect of this filtering on the combined motion fields, which Figure 4 is relating to.*

Page 8, Lines 20/21: "combined motion fields" --> How about the retrieval success of satellite derived ice motion during summer? Aren't during summer these "combined motion fields" only based on buoy motion and NCEP/NCAR winds? Please be more specific in your description. Perhaps you adopted the method of Kwok and use the 19 GHz channels now as well?

*This section deals with reanalysis winds, so further discussion of the combined fields does not fit here. Therefore, we have added text to Section 2.2 noting the issues of summer retrievals. We also make clear there that PM motions are derived through the summer even though they have higher errors; there are also much fewer valid vectors during summer. We also reference Kwok's use of the 19 GHz channels from AMSR-E.*

Page 12, lines 11-24:

I note that the description of how C and D are found is vague. Nobody could re-do this analysis. It is not clear - after your previous statements about the influence spatial resolution appears to have on precision (or accuracy) - why C is set to a constant value when you are operating with individual ice motion resolutions between ~25 km (using SMMR/SSMI/SSMIS 37 GHz), 12.5 km (same but using near-90 GHz or AMSR-E near-37 GHz), 6.25 km (AMSR-E 89 GHz) or 4 km AVHRR. Please try to be more specific in your description.

*We've added details to the descriptions of C and D. We note that a constant C value for all of the satellite-sources is sub-optimal. This is certainly something that can be improved for a future version. But as this paper describes Version 4 where these values were not change*

Also, later on you are referring to the improved interpolation used in version 4, pointing out that the buoy weight drops to zero (e.g. Page 20, Line 2) at a certain distance to the buoy. However, it is not clear what this distance is, whether we speak about 50 km, 500 km, 2000 km of these, presumably, correlation lengths. It is not clear how these were calculated and whether and how much these differ between the single products and input data used as input for the merged product.

*We have added the value of D. We also edited the text on the buoy weight dropping to zero as that was not correct. The buoy weight doesn't necessarily drop to zero. Either the buoy distance falls outside of the range (D) of a given point (Version 3) or the weight falls below the top 15 weights of observations surrounding a given point (Version 4). In other words, the buoy weight doesn't drop to zero, rather buoy estimate falls outside of the criteria for inclusion at a given point.*

Page 13, Lines 5-9:
I strongly recommend to provide more details here. While your Figure 3 illustrates nicely the effect of this amendment from version 3 to version 4 what is missing is information about the distance within which these 15 highest-weighted ice motion vectors are selected. I assume it has something to do with the correlation length scales (please provide examples of these). If not, then one could provocatively say that at any point within the Arctic Ocean the gridded ice motion product is solely determined by buoy motions if at least 15 buoys are reporting - because, as I understand your averaging with C=0.95, these 15 buoys would have the highest weight and would be used no matter how far away they are from the grid cell considered.

*We've added text to clarify this in the text. It is possible that 15 buoys may be the 15 highest weighted if the buoys are close enough to the interpolated point. However, as Equation 1 indicates, the weight falls off exponentially with distance. So, buoys that are farther away will be less weighted than nearer wind or satellite estimates and they will fall out of the "top 15". Thus, it is not true that buoys will always be included no matter how far away they are.*

Page 14, Lines 18-27:
- Please add information whether you used the daily or the weekly gridded product.

*Daily. This is added to the text that has been moved to the supplement.*

- Please add the fraction of discarded CRREL buoy motion estimates; I assume it is very small.

*We have added this number to the supplement. Yes, it is small.*

- Please provide a map of the tracks of these buoys for illustration. In light of the next paragraph you should highlight the track the year 2015 (unless you color code them anyways and unless you decide to use all years instead of just one).

*We've added a map all of the buoy tracks for 2015 in the supplement, which is the focus year for the wind-sensitivity study and also the year with the most buoy observations. We feel adding other years would unnecessarily clutter the map.*

- Please provide information about the processing steps. The combined gridded ice motion product has u- and v-component of the ice motion aligned with the EASE grid. How about the CRREL buoy data? How did you compute the u- and v-components of the CRREL buoy motion in the EASE grid? This description should also include a notion about how many CRREL buoy position observations of one day form one daily estimate.

*We've added these details in the supplement.*

- Please describe in detail how you compared the data and hence how you ended up with the numbers shown in Table 3, i.e. did you compare absolute values or the "native" positive and negative values? I assume you did the latter. Did you also compare the direction and the absolute value of the ice motion vectors?

*It was the native positive and negative values – this yields the bias values. We compared u-component and v-component (relative to EASE grid), not speed and direction. We've added this information to the supplement text.*

- Please provide at least one plot for each ice-motion component which illustrates how the ice-motion values scatter. You could do this as a scatter-plot, a 2-dimensional histogram, regular histograms, ... whatever you like, but please show the reader / user more than just a table.

*We have added 2-dimensional histograms into the supplement that we have added. We've chosen to add a supplement to address this suggestion because the main manuscript is already quite long. Table 3 shows that Version 4 is improved vs. Version 3. The supplement provides further details supporting this conclusion.*

Page 18, Lines 19-27:
- Just to re-cap: The 4X oversampling is something introduced from version 2 to version 3? Or was this already introduced in version 1?

*No. The 4X oversampling was used from the very beginning of the method and thus was in Version 1 of the product.*

- Line 25: I don't understand why the different temporal sampling of SMMR explains why the difference in the weekly average drift speed between version 3 and version 4 is near zero.

*This was not originally explained well and we have edited the text to be clearer and more complete. The SMMR data were not changed at all between Version 3 and Version 4. The only change in the combined motions during the SMMR era was the changing in the weighting to use the 15 closest vectors. Since there are relatively fewer buoys, and there are not generally a lot of AVHRR estimates, the effect is small. The over-filtering issue affected SSMI and SSMIS, which were reprocessed for V4; so, the change to V4 is bigger for this period. AMSR-E was not affected by the over-filtering and also was not reprocessed for V4; thus, the contribution of AMSR-E motions muted the over-filtering of SSMI-SSMIS and reduced the V4-V3 difference. The difference increased again after AMSR-E dropped off the record and the differences increased again.*

Page 19, Lines 1-3: How much of the larger differences between version 3 and 4 during summer can be attributed to the change in weights in combination to a predominant usage of NCEP/NCAR winds based ice motion [assuming that in summer there as substantially fewer valid PMW ice motion vectors]?

*Yes, there are fewer PM vectors during summer, so winds are used relatively more. With winds having a lower weight, the buoys will be used over a relatively larger distance compared to PM. The other factor is the change to address the over-filtering in SSMI/SSMIS. The relative effects can be seen in Figure 7 where the seasonality is larger in the SSMI/SSMIS-only period where the over-filtering has an effect vs. the SMMR period where only the weighting is an effect and the AMSR-E period where the over-filtering of SSMI-SSMIS is muted by the presence of AMSR-E motions. We've added text to explain this.*

+++++++++++++++++++++++++++++++++++++++++++++++++++++++++++++++++++++++

Editoral comments / typos:
Page 4, Line 11:
I'd say Szanyi et al. (2016) referred to the sea-ice motion product as well; therefore I suggest to add "sea-ice motion" here as well. This would also comply better to the "both products" notion in the next line.

*Added "motion".*

Page 4, Line 21:
"monthly" --> It might make sense to either correct the NSIDC web page or the text. Monthly estimates seem not to be available.

*Removed "monthly". Monthly fields are no longer part of the product. We've added this note to Table 1 as well.*

Page 5, Line 19/20: "a region around that grid cell" --> Does this mean that the search window is

centered at the center of the grid cell center and hence extends by typically 25 km in x- and y-direction?
"typically 50km" --> this reads as if different search window sizes are used. Is this the case? Please be more specific.

*The 50 km is not from the center of the grid cell, but from the edges of the grid cell – i.e., for a 25 km grid cell, it would be two grid cells in each direction. We clarified this in the text.*

Page 5, Lines 22-24: "The highest correlation value, i.e., the correlation peak, is determined to be the offset in the position of the grid cell between the earlier and the later image; then, the ice velocity is computed by dividing this offset by the time separation between images."
I don't think that the correlation itself gives the spatial offset, does it? How about "The highest correlation value, i.e., the correlation peak, is assumed to coincide with the most likely offset in the position of the grid cell between the earlier and the later image. This offset in the position yields a displacement vector pointing into the direction of the ice motion while the ice velocity is computed by dividing its magnitude by the time separation between the two images used."

*We changed the wording as suggested.*

Page 6, Line 11:
"daily": SMMR did not provide daily data. This needs to be corrected in the text.

*Corrected.*

Page 6, Line 14:
"a gridded a resolution" --> "a gridded resolution"

*Corrected.*

Page 6, Line 15:
"many similar" --> please be more specific and refer to these, e.g., with a reference. Perhaps "many" could be deleted?

*References added and "many" deleted.*

Page 6, Line 20:
What about SMMR? Is the oversampling applied here as well?

*Yes. We note in the text that the 4X oversampling is applied to all satellite estimates.*

Page 6, Line 21: I suggest to add "theoretical" to "motion precision".

*Done.*

Page 6, Paragraph starting at line 25:
- Why were AMSR-E data not used for the ice motion product of the Southern Hemisphere?

Please either add the reasoning or at least mention that AMSR-E was only used in the Northern Hemisphere.

*We added a note to the text that AMSR-E is only used in the Northern Hemisphere.*

*This was a decision made at the time. In the original product development, the Southern Hemisphere was not a particular focus and the motion fields were developed at a basic level. This is something that we will look into further – adding AVHRR, AMSR-E, and wind motions to the Antarctic fields.*

- This part: "AMSR-E had more than double the spatial resolution of the previous sensor, 6.25 km gridded resolution for some channels, so its motion resolution was likewise improved. So, during this period (2002-2011), it was also used as a source for ice motions." should be re-written as it does not read well. It might help to refer to Table 1 by the way. "of the previous sensor" --> better "than SSM/I and SSMIS"

*This has been rewritten to read better.*

- On Page 6, Lines 7 and 9, you provide references for the SSM/I - SSMIS data product and the SMMR data product. An adequate reference for the AMSR-E data is missing.

*AMSR-E references added.*

Page 7, Lines 6-14:
- Why were AVHRR data not used for the ice motion product of the Southern Hemisphere? Please either add the reasoning or at least mention that AVHRR data were only used in the Northern Hemisphere.

*Similar to AMSR-E above. This was a decision made at the time. In the original product development, the Southern Hemisphere was not a particular focus and the motion fields were developed at a basic level. This is something that we will look into further – adding AVHRR, AMSR-E, and wind motions to the Antarctic fields.*

- Your reasoning that AVHRR data were not used after the year 2000 because AMSR-E became available reads a bit strange given the fact that AMSR-E became available in May 2002. I assume that AVHRR data usage was simply confined to version 1 (and version 2) of the NSIDC sea-ice motion product and/or that the AVHRR data set used those days (...) simply terminates at the end of 2000?

*The AVHRR product used ended in 2000. We added this note to the text. While other AVHRR products could be used to continue use of AVHRR, particularly for 2001 and 2002 before AMSR-E's launch, it was deemed not worth the effort at the time. In a future version, we plan to look at newer AVHRR products that continue past 2000 and include them in the product.*

- AVHRR has visible and infrared channels as you state. But which are used for the "Daily

gridded composites"? In other words, are the AVHRR ice motion vectors based on visible or infrared data?

*Both visible and infrared are used – visible during summer, infrared in winter.*

- "higher resolution = more accurate motion estimates" --> Is this the case? Or is it mainly the precision which improves as resolution refines? In any case, it might be superb to add information about an inter-comparison between buoy, AVHRR and, e.g., SSM/I based ice motion estimates illustrating this statement in the supplementary material.

*We changed "accurate" to "precise".*

- Line 12: "(6.25 km vs. 4 km)" This reads as if the often strongly weather influenced near-90 GHz channels are the backbone of the ice motion estimates using AMSR-E. Is this the case? If not, then you need to mention the 12.5 km grid resolution for the near-37 GHz data. If yes, then your description about the over-sampling of the 25 km gridded data (near 37 GHz data) is given a bit too much weight because then one would assume that also for SSM/I and SSMIS the backbone data set are the near-90 GHz channels which in fact come at 12.5 km grid resolution. Your writing is hence inconsistent and should become more specific.

*Here we make the point that AMSR-E near-90 GHz have a similar resolution as AVHRR, but can see through clouds. So AVHRR largely duplicates the AMSR-E near-90 GHz capabilities in terms of spatial resolution, but with so few vectors that with AMSR-E there is even less contribution from AVHRR. Thus, use of AVHRR was discontinued.*

*We have edited this paragraph to make this clearer. And we also make it clear that the 36/37 GHz channels of AMSR-E are also used. We also added text in the SSMI-SSMIS discussion above that both the 37 GHz and 85 GHz channels are also used.*

- On Page 6, Lines 7 and 9, you provide references for the SSM/I - SSMIS data product and the SMMR data product. An adequate reference for the AVHRR data is missing.

*We added this reference for the AVHRR data:*

*W. Emery, C. Fowler, T. Haran, J. Key, J. Maslanik, T. Scambos 2000. AVHRR Polar Pathfinder Twice-Daily 5 km EASE-Grid Composites, Version 3. Boulder, Colorado USA. NSIDC: National Snow and Ice Data Center. doi: https://doi.org/10.5067/HRMXN6PE1Q0Q.*

Page 10, line 16: I suggest to add "theoretical" in front of "limit of precision"

*Added.*

Page 10, paragraph starting on Line 16:
- This is a very globally written, non-specific paragraph. Please note clearly which versions of the NSIDC ice-motion products were compared here. It would also be important to know whether these comparisons were done using the single ice-motion vectors or using the daily

gridded product. It would further be important to learn about the amount of data compared here, i.e. are we talking about several years' worth of data or a few months or even only days? Example 1: In Line 21 it only states "the Lagrangian motion product". Example 2: In Line 23 it is not clear whether these "SSMI-derived daily velocity components" are from a gridded product or single motion vectors. The same applies to Line 24, wherein you write about the AMSR-E ice motion. Here, one asks oneself what "error" is meaning in particular. Example 3: In Lines 25/26 it only says "the ice motion data". Neither the time period, nor the gridding or the version number are given. Note: a high correlation is wonderful but how about bias and RMS error for the Sumata et al. studies?

*This section is meant to provide an assessment of the general error characteristics of the source motion estimates based on previous studies. We've renamed the section title to be clearer, and made clarifying edits within the section. We removed "Lagrangian" and refer simply to "SSMI-derived", which is what was analyzed in Kwok et al. (1998). We added clarifying text in the AMSR-E sentence. We removed the sentence about the Sumata comparison because that was for the gridded composite product, which isn't relevant for this section.*

Page 11, Lines 5/6: Please check this first sentence. It gives not clear meaning.

*Edited the first sentence to be clearer.*

Page 12: Line 2 "and monthly" --> should be deleted as there appears to be no monthly ice motion field available (anymore). At least they are not accessible via the product's web page.

*"and monthly" removed.*

Page 12, Lines 11-24:
- You mention three times that C is a assigned a value of 0.95 for buoys. One time might be enough.

*The paragraph was rewritten to be less repetitive.*

Page 13, Line 14: Please provide a reference for the correlation length scale of ice motion. Is it the same in the Arctic?

*Added Meier et al. (2000) as a reference.*

Page 13, Line 20: The perfect place to refer back to Figure 2 which nicely illustrates how smooth the "Winds" ice motion is compared to the PMW ice motion.

*We added a note here that the winds fill in for PMW in the Arctic, which ameliorates the PMW over-filtering; we also add a reference back to Figure 2.*

Page 13, Line 21: "corrects this over-filtering" --> How? This is your recent work and should be detailed more. See my previous comment on this issue.

*We've added text to explain this in more detail.*

Page 14, Lines 1-15:
- I note that in this paragraph finally you do not mention the monthly product anymore.

*Thank you. Old habits are hard to break.☺*

- Line 10: "discretization effects" --> Could these also be caused by the fact that your 4X downscaling applied to original 25 km gridded resolution (with even coarser footprint) data often cannot resolve the anticipated smaller-scale variations in ice motion but in contrast enhances noise - particularly in the direction of the motion? I recommend to spend a sentence or two about this issue and also take into account the paper by Lavergne et al. (2010).

*The discretization effect we mention here is akin to the quantization noise noted in Lavergne et al. (2010) – i.e., the MCC can only estimate if there is a displacement in one grid increments: a parcel is estimated to move only 0 km, 25 km, 50 km, etc. The 4X oversampling cuts this by a factor of 4 – e.g., the parcel is detected to move 0 km, 6.25 km, 12.5 km, 18.75 km, 25 km, etc.*

*There may be more noise in the 4X fields because it allows for more variability, some of which may be an artifact of the oversampling. We've added text to provide more detail of the 4X oversampling and we reference Lavergne et al. (2010).*

Page 15, Lines 1-13:
- In Line 1 please refer to the respective paragraph or subsection.

*Added "in Section 2 (Reanalysis winds)".*

- Line 3: Is the ice speed increasing? Then we are talking about a positive trend and not about an "increasing trend". Or is an already existing positive (or negative) trend increasing? Please be more clear in your formulation.

*We changed "increasing" to "positive".*

- What is your explanation for the 10% difference between versions 3 and 4 in the bias of the v-component compared to the near-1 % bias of the u-component?

*This is a good question. The main reason is that while the u-motion tends to be fairly equal in each direction, the v-direction is largely negative (to the bottom of the grid) because the way the grid is oriented, the TDS and the Fram outflow are almost totally in the negative v-direction, with little u-component. Also, the northern half of the Beaufort Gyre is also in the negative v-direction. Only the southern Beaufort Gyre is primarily +v. So, I think because v is so skewed, there is more apparent bias.*

- Lines 11-13: Frankly speaking I am very surprised about the small impact of using 2% instead of 1% to derive ice motion from NCEP/NCAR winds, but I am quite confident that this can be explained with your choice of data. Therefore, while I appreciate your future plans with respect

to regions and long-term trends I am suggesting at first to carry out the comparison for the entire CRREL buoy motion data set (why 2015?) available (as used in the previous paragraph). Secondly, as you correctly write, the weights used for NCEP/NCAR wind-speed based ice motion are very low - particularly during winter when you have plenty of useful PMW ice-motion estimates. I am sure this changes during summer melt. Therefore it would certainly be much more informative to show the comparison with CRREL buoy ice-motion data alongside with purely NCEP/NCAR based ice motions (i.e. those exemplified in Figure 2 c). If you do this as a time-series you could also account for the fact that during summer NCEP/NCAR winds-based ice-motion estimates potentially play a substantially larger role for the combined gridded ice motion product.

*We chose 2015 simply because it was a year with a high number of buoys and the last full year available in the CRREL archive. As we're interested in a change from 1% to 2%, any reasonable number of samples (i.e., large enough to have reasonable statistics) will suffice.*

*While it seems like doubling the winds might have a bigger effect, it demonstrates that the weighting of the winds is small compared to the passive microwave and particularly the buoy motions.*

*The rationale of the comparison presented here is to investigate the effect of the wind-ice scaling (1% vs. 2%) on the combined gridded motions. It is not to analyze the relationship of the winds to the buoys; while that is an interesting study, it is tangential to the purpose of this paper.*

Page 15, Line 15: "the motion product" --> which one? The combined gridded one? Please write specifically what you use and do.

*Yes, the combined gridded one. We have clarified this.*

Page 16, Line 7: "1" --> "one"

*Changed.*

Page 16, Line 12: Is an "increment" by definition positive?

*Yes, the parcels get one year older. We changed "incremented" to "increased" to be clear.*

Page 18, Line 17: "motion age"?

*Fixed.*

Page 18, Lines 20-21: "motions are smoothed out"? So they are zero?

*No, they are not zero. This refers to the temporal averaging the "smooths out" daily variability. We have rewritten this to be clearer.*

Page 19, Line 7: "increasing trend" <--> "positive trend"; see my previous comment.

*Changed.*

Page 19, Lines 7/8: Isn't it strange to see that for the version 4 product, which is faster than the version 3 product especially for the SSM/I period (see Figure 7) you get a larger positive trend from version 4 than from version 3 data? Perhaps you should mention in the context of Figure 7, that after the AMSR-E period, the difference between versions 3 and 4 is even larger than before the AMSR-E period.

*We've added text earlier to more thoroughly describe the characteristics of the V4-V3 differences in Figure 7 and here we note that these differences do impact the trend values.*

Page 20, Line 11--: What does, to your opinion, explain the fact that the differences between version 4 and version 3 are near zero for all ice-age classes for winters 2005/06 'til 2012/13? Do we really understand why for these winters (dominated by AMSR-E 89GHz PMW ice motion input at 6.25 km grid resolution) the change in the interpolation method between version 3 and version 4 appears to have no influence?

*This is because during these years, AMSR-E dominated so the filtering change to SSMI/SSMIS had minimal effect.*

Page 21, Line 8: Please add the information that the winds originate from the NCEP/NCAR atmospheric reanalysis product.

*Done.*

Page 21, Line 28: I suggest to replace the ESA CPOM 2015 link by the recent paper by Salilla et al. from 2019 in The Cryosphere; this paper gives a comprehensive overview about currently existing CryoSat-2 sea-ice thickness products.

*Replaced reference.*

Page 22, Line 1: "ice age motion" ?

*Corrected.*

Page 22, Line 6: Please add "like suggested by Korosov et al. (2018)" behind "... EASE grid cell" to indicate that this is a feasible idea which has been followed up with already by other scientists.

*Done.*

Figure 1: I note that this figure omits the AVHRR data.

*We show this figure as example of the inputs and the interpolated field. We chose a relatively recent year (2016) as being of more interest. Other years could also include AVHRR or AMSR-E, but a 4-panel figure is easier to read and gets the main point across – showing the individual*

*source motions and the combined, interpolated field side-by-side. We have added a note to the caption mentioning AMSR-E and AVHRR.*

Figure 3:
I note that in panel (a) a lot of the circular features with substantially different ice motion than the surrounding do not contain a red dot for a buoy being present. One example is found in the northern East Siberian Sea and another, more pronounced example is north of the Laptev Sea. How do you explain this?

*We double-checked this and all buoys are marked with the red dot. The other regions are SSMI estimates that have noticeable differences than the wind-driven motion estimates. Like the buoys, there is a drop off as the higher-weighted SSMI drop out of the 15 nearest (V3) or highest weighted (V4) ranking.*

- Is the grey scale the same between panel (a) and (b)? I am asking because I am surprised to see that the area with high positive values in the northern East Siberian Sea has so much increased (both in extent as well as in magnitude of the values) from panel (a) to panel (b). Please comment on this.

*Yes, the scale is the same. The high positive values in the northern East Siberian Sea is because there is one buoy near the ice edge and no other buoy to the east. So, using the highest weighted approach for V4, that buoy's influence extends out much farther to the east and northeast. There isn't a similar change to the west because there is another buoy just to the west that "mutes" the influence of the eastern buoy in the region.*

- I recommend to also show the respective uncertainty information for these two maps to illustrate to the user what the effect of this step is on this parameter. This might be an important information for users attempting to assimilate your product.

*We've added information on the uncertainty field to the Figure 3 discuss. We have decided to not add a figure with uncertainty because it shows much the same pattern as in Figure 3 and thus in our view does not add information.*

Figure 7: Please state in the text and the Figure caption on which grid cells this difference is computed. Did you use all grid cells? This could mean that you are superposing the true difference between version 3 and version 4 ice motions with an influence of the change in the ice drift distribution due to the sea-ice retreat which (the influence) might be different for version 3 than for version 4. Therefore I ask: How would this figure look like if you limit this comparison to the NSIDC region "Arctic Ocean"? Would the differences still be of the same magnitude?

*Yes, we used all valid motion values. There is no difference between the sea ice mask in Version 3 and 4. It is true that different regimes will have different motion characteristics and these may vary spatially and temporally. This may be one reason why there is seasonality in the difference – in summer, motions are only in the Arctic Ocean, and there is more free drift. It is an interesting question as to how these motions vary seasonally, but we feel this is beyond the scope of this manuscript. The figure is included to show how the motions from V4 and V3 differ.*

Figure 9 and its interpretation:
- What is special in winters 1995/96 and 1996/97 causing version 4 and version 3 4+year old ice extent to be similar even though we are still in the middle of the SSM/I period?

*This is indeed an interesting feature. It may be related to the high variation between the 1995 summer minimum (record low to that point) and 1996 (very high). It could also be related to the end of the high AO period around that time. It is something we'd like to look into further, but is beyond the scope of this paper.*

- While the previous issue is difficult to understand the one I am refering to now is logical and should be explained in more detail in your text. Apparently, introducing AMSR-E data in summer 2002 did not have an impact on the 4+ years old ice immediately but it took until winter 2004/05 or 2005/06 to see its effect on the difference between both product versions. Likewise, the termination of AMSR-E usage in fall 2011 and hence switch to coarser resolution SSMIS data manifests as late as 2014/2015. This is logical because it takes 3 years for the benefit of first the finer, later the coarser resolution to have an effect on the old ice.

*We added a phrase to the discussion noting that age is gradually affected over the years when new data sources come in.*

Figure 10:
- Please provide a similar time axis as you used in Figure 9.
*Done.*

Figure 11:
- Please provide a similar time axis as you used in Figure 9.
*Done.*

- What is the rationale behind including Kara and Barents Sea into this plot?

*This is the region of the NSIDC "Arctic Ocean" mask. The rationale is that it includes all areas in our domain that may have a reasonable amount of MYI that could circulate into the Arctic. Hudson and Baffin have only minimal amounts of MYI and they melt out each summer. There are no data in the Canadian Archipelago. Bering Sea and Okhotsk have only FYI. Greenland Sea has MYI, but it quickly drifts south and melts. Barents and Kara generally have minimal MYI, but what it has can potentially circulate into the main part of the Arctic, so those regions are included. This region has been used in past reporting of age values, notably the Arctic Report Card. We've added a reference to the latest report card here.*

- Caption: I have a problem with usage of the word "trend" here. To me a trend is something I compute from a time series of data, e.g. via a functional relation, and it has, in its simplest form an intercept and a slope. What you plot here are, to my opinion, time series of the fractions of the different ice-age classes.
*We changed "trend" to "timeseries".*

[revised manuscript text omitted]
 2000-2016. All of the CRREL buoy positions were converted to EASE grid coordinates and the u-component and v-component of velocity (relative to the EASE grid) was derived from the change in position over a 24-hour period. The combined estimate closest to each CRREL buoy

20   was selected for comparison; thus, each comparison was made generally within ~25 km. A small number (<0.1%) of CRREL observations with erroneous velocities that were obviously too large were removed from the comparisons. This resulted in a total of nearly 26,000 pairs of observations from buoys and the combined motion field. The results (Table S2) show that biases are around -0.1 cm/s for the u-component and around -0.66 to -0.69 cm/s for the v-component. Most notably, the biases were slightly reduced in

25   Version 4, indicating that the improvements in processing do result in improved accuracy of the motions. Similarly, the error standard deviations are around 4 cm/s for both velocity components and Version 4 reduces this error by ~0.3 cm/s over Version 3.

**Commented [Walt1]:** This is retained in the tracked-changes version to show edits made in response to reviewer comments. In the final draft, this section will be cut from this section as the material has been moved to the Supplement. Some further editing is done in the Supplement to integrate it with new material in response to the reviewers' comments.

[revised manuscript text omitted]